# GANDALF: GENERATIVE ATTENTION BASED DATA AUGMENTATION AND PREDICTIVE MODELING FRAMEWORK FOR PERSONALIZED CANCER TREATMENT

**Aishwarya Jayagopal[1], Yanrong Zhang[1], Robert John Walsh[2], Tuan Zea Tan[3],**
**Anand D Jeyasekharan[3], Vaibhav Rajan[1]**
[1] School of Computing, National University of Singapore
[2] National University Cancer Institute, Singapore [3] Cancer Science Institute of Singapore
{e0674492, e1068520}@u.nus.edu, {robert_walsh}@nuhs.edu.sg,
{csittz, csiadj, vaibhav.rajan}@nus.edu.sg

## ABSTRACT

Effective treatment of cancer is a major challenge faced by healthcare providers, due to the highly individualized nature of patient responses to treatment. This is caused by the heterogeneity seen in cancer-causing alterations (*mutations*) across patient genomes. Limited availability of response data in patients makes it difficult to train personalized treatment recommendation models on mutations from clinical genomic sequencing reports. Prior methods tackle this by utilising larger, labelled pre-clinical laboratory datasets ('cell lines'), via transfer learning. These methods augment patient data by learning a shared, domain-invariant representation, between the cell line and patient domains, which is then used to train a downstream drug response prediction (DRP) model. This approach augments data in the shared space but fails to model patient-specific characteristics, which have a strong influence on their drug response. We propose a novel generative attention-based data augmentation and predictive modeling framework, GANDALF, to tackle this crucial shortcoming of prior methods. GANDALF not only augments patient genomic data directly, but also accounts for its domain-specific characteristics. GANDALF outperforms state-of-the-art DRP models on publicly available patient datasets and emerges as the front-runner amongst SOTA cancer DRP models.

## 1 INTRODUCTION

Cancer, a leading cause of deaths worldwide (Dattani et al., 2023), imposes a significant burden on global healthcare systems (Lopes, 2023). It is caused due to the presence of alterations (*mutations*) in the human genome, resulting in uncontrolled replication of cancer cells. Cancer patients exhibit a great deal of heterogeneity in their genomic mutation profiles, even when they have the same cancer type. This heterogeneity causes patients, of the same cancer type, to respond differently to the same treatment (Liao et al., 2023), making cancer treatment challenging (Wahida et al., 2023). Treatment, today, is largely guideline-based and prescribes drugs based on the cancer type (Planchard et al., 2018; Conroy et al., 2023; Morris et al., 2023). This approach fails to account for heterogeneity in patient mutations, and its impact on treatment outcomes. Precision oncology (Sosinsky et al., 2024; Collins & Varmus, 2015) is gradually shifting focus from a "one-size-fits-all" approach to more personalized treatment strategies.

To aid precision oncology, cancer patients undergo genomic sequencing as part of clinical diagnostics (Colomer et al., 2023). Clinical sequencing panels (Milbury et al., 2022; Wei et al., 2022) identify the set of mutations present in specific sections of the human genome (called *genes*), which have a known association with cancer. Cancer patients can exhibit a varying number of mutations in each of these genes (Saito et al., 2021). These mutations interact with each other and the drug in complex ways to determine patient response to treatment (Liu et al., 2022). While clinical trials have identified drugs that target specific mutations, these studies have largely been restricted to single mutations (Brachova et al., 2013; Randic et al., 2023). Conducting large scale clinical trials

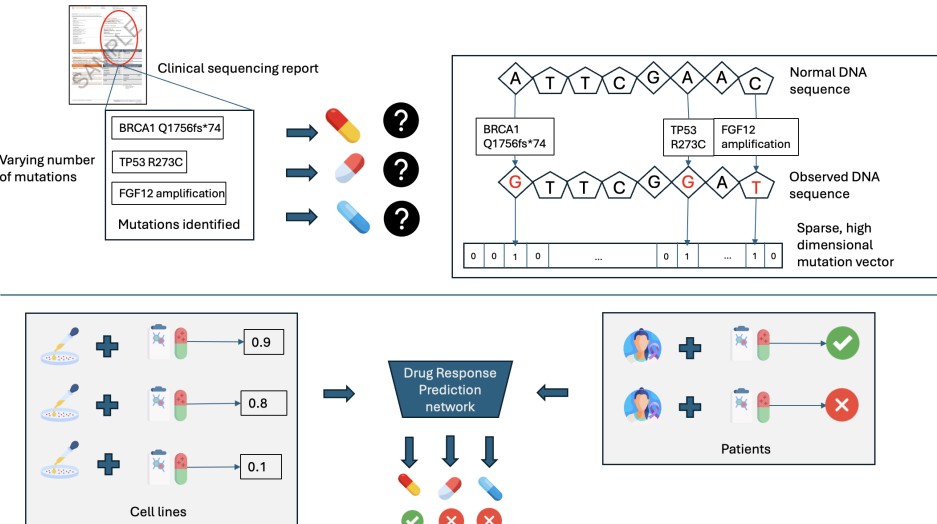

Figure 1: Overview of clinical challenge in cancer drug response prediction.

for all possible combinations of mutations in $\sim 20000$ genes of the human genome is practically intractable, thereby limiting their ability to identify the right treatment when a patient exhibits multiple mutations.

Machine learning (ML) approaches provide a promising avenue to predict patient response $y_p$ to drugs $d_p$, based on the set of mutations $X_p$ in their genomic profiles. However, guideline-based treatment in clinics prescribe only a small subset of drugs from all drugs approved for clinical use, thereby limiting the availability of labelled patient data $(X_p, d_p, y_p)$. The resulting scarcity poses a significant challenge in training supervised ML models to predict drug response in patients. Prior methods in Drug Response Prediction (DRP) literature have tackled this using data from a related domain called "cell lines". Cell lines (Ghandi et al., 2019) are cancer cells extracted from patients, which are then cloned under controlled laboratory settings. Each clone $X_c$ is administered a different drug $d_c$, and the corresponding response $y_c(X_c, d_c)$ is measured for various drug concentrations. Since these cells are studied outside the human body, it is possible to obtain $y_c$ for a large set of drugs $\mathcal{D}$, resulting in abundant labelled data.

However, models trained only on $(X_c, d_c, y_c)$ do not work well on patients (Mourragui et al., 2019; 2021; Sharifi-Noghabi et al., 2020). This is attributed to the inherent differences between patients and cell lines. As cell lines are studied outside the human body in the absence of blood vessels and the immune system (called *tumor microenvironment*), these cells can acquire mutations differently compared to patients, i.e. $P(X_c) \neq P(X_p)$. In addition, $y_c \in [0, 1]$ depends on drug concentration and number of surviving cells (called Area Under the Dose Response Curve, AUDRC), while $y_p \in \{0, 1\}$ indicates good or bad response (called Response Evaluation Criteria in Solid Tumors, RECIST, based on change in tumor volume), i.e. $domain(y_c) \neq domain(y_p)$, as shown in Figure 1.

Prior DRP methods (Jayagopal et al., 2024; 2023; Kim et al., 2024; He et al., 2022) have addressed these differences by learning shared domain-invariant representations $Z_s$ between $X_c$ and $X_p$, which are then used to train a downstream drug response prediction network $f$. Transforming $X_c$ to $Z_s$ increases samples in the shared space and allows $f$ to use the larger $(Z_s, d_c, y_c)$ in training, thereby tackling the data scarcity issue. However, $Z_s$ does not capture patient-specific characteristics in $X_p$, which can influence $y_p$ (Liao et al., 2023; Zhai & Liu, 2024). To capture this, we need to augment $X_p$ directly. Prior DRP methods, except WISER, neglect this. WISER (Shubham et al., 2024) performs data augmentation by pseudolabelling unlabelled patient profiles $X_{p(u)}$ using $(X_c, d_c, y_c)$ and trains $f$ by combining $(X_c, d_c, y_c)$ and pseudolabelled $X_{p(u)}$. However, while combining the two datasets, WISER assumes $domain(y_c) = domain(y_p)$, and does not account for $P(X_{p(u)}) \neq P(X_c)$. We tackle these issues using **GANDALF**, a *Generative AttentioN based Data Augmentation and predictive modeLing Framework*. GANDALF augments $X_p$ directly, by generating more "patient-like" samples $X_{aug}$ leveraging available $X_c$. It also generates their response labels $y_{aug}$ to drugs $d_{aug} \in \mathcal{D}$. Unlike WISER, it explicitly models $domain(y_c) \neq domain(y_p)$ and $P(X_p) \neq P(X_c)$.

Data augmentation strategies are known to improve prediction performance in various fields of ML, like computer vision (Khosla & Saini, 2020) and natural language processing (Shorten & Khosh-goftaar, 2019). This is usually achieved through data transformations where identifying the label of the transformed data is relatively easy, e.g., a rotated image of a dog retains the label 'dog' after transformation. However, it is difficult to find such 'label-invariant' transformations for genomic data (Lacan et al., 2023). Although genomic data can be augmented by interpolation of available samples or sampling new data points from a known distribution, assigning labels to these samples is difficult. Data points, which may be "close" together in the representation space, can still exhibit different responses to drugs. If patients are represented by binary vectors (each element corresponding to a gene, 1 indicating presence of mutations in a gene and 0 the absence), a perturbation is equivalent to addition or removal of a mutation. This perturbation can impact the functioning of the cells and the response to treatment (Hale et al., 2024). Identifying the response associated with each perturbation is difficult due to scarcity of labelled data, making data augmentation strategies challenging in DRP.

Though conclusively identifying labels for all possible perturbations is still an open problem, GANDALF takes a step towards leveraging data augmentation in DRP, by utilising available labelled data from cell lines. It generates $X_{aug}$ by transforming $X_c$ and assigns $y_{aug}$ for generated $(X_{aug}, d_{aug}), d_{aug} \in \mathcal{D}$ by leveraging labelled information from $(X_c, d_c, y_c)$. We use attention mechanisms to ensure that $X_{aug}$ retains information from $X_c$. $(X_{aug}, d_{aug}, y_{aug})$ is then used with $(X_p, d_p, y_p)$ to train a downstream DRP classifier. Our paper makes the following contributions:

- We are the first to tackle, through a novel data augmentation approach, the challenging problem of limited labels for sparse patient genomic data, in cancer drug response prediction.
- We propose GANDALF, a generative, semi-supervised, attention-based data augmentation framework which uses labelled samples from the related cell line domain to generate labelled patient data.
- GANDALF performs data augmentation through a novel synthesis of denoising diffusion probabilistic models, transformers and multi-task learning.
- GANDALF demonstrates an improvement of upto 10.96% over SOTA DRP methods, in predicting patient response to drugs, on key benchmark datasets comprising real patient samples with responses to clinically approved anti-cancer drugs. GANDALF also outperforms baseline genomic data augmentation and pseudo-labeling strategies by 21% and 2.5% respectively.

## 2 RELATED WORK

### 2.1 DRUG RESPONSE PREDICTION MODELS

Prior DRP models perform transfer learning between the source domain (cell lines) and target domain (patients). These methods can be inductive, transductive or unsupervised (Pan & Yang, 2009), based on their use of labelled patient data. Inductive methods, like AITL (Sharifi-Noghabi et al., 2020), drug2tme (Zhai & Liu, 2024) and TCRP (Ma et al., 2021) use both labeled cell line and patient samples. They may either use multi-task learning approaches or few shot learning to capture the differences in label distribution across the two domains. Transductive methods like TUGDA (Peres da Silva et al., 2021), WISER (Shubham et al., 2024), PANCDR (Kim et al., 2024) use labeled cell line and unlabeled patient samples. The unjustified assumption is that the response label does not change across the domains. To this end, most papers convert the continuous valued cell line response to discrete categories as seen in patients, using arbitrary thresholds. Few methods, like CODE-AE (He et al., 2022), rely on unsupervised transfer learning using unlabeled cell line and patient datasets in pre-training. However, in most cases, the goal was to learn a shared representation space between the domains. The shared representation was then used to train a downstream DRP model. While the shared representation captures the similarities across the domains, this approach largely neglects the patient-specific characteristics, which is relevant for drug response prediction.

### 2.2 GENOMIC DATA AUGMENTATION

Genomic data augmentation is difficult due to lack of known label-invariant transforms (Lacan et al., 2023). Most existing methods augment transcriptomic data (Das & Shi, 2022; Chen et al., 2020), which is unavailable in a clinical setting. A few recent methods (Yu et al., 2024; Lee et al., 2023;

Duncan et al., 2024; Lee et al., 2024) have augmented mutations, but they assume that the biological function and associated labels do not undergo changes during data transformation. Moreover, none of these methods focus on cancer drug response prediction as the downstream task, where it is known that even the addition or removal of a mutation can cause a change in drug response (Liao et al., 2023). Thus, patient mutation data augmentation for cancer drug response prediction is an open problem. GANDALF proposes a way forward, by using prior information available in labelled cell lines to augment patient mutation data and to generate associated labels for DRP, rather than assuming label invariance.

## 3 METHOD

### 3.1 PROBLEM FORMULATION

Given a patient genomic mutation profile $X_p$ and drug $d_k$, the goal in drug response prediction (DRP) is to classify whether the patient would respond well (label $y_p = 1$) or not (label $y_p = 0$), i.e. to learn a classifier $f_{d_k}(X_p) : \mathcal{R} \rightarrow \{0, 1\}$. Let $\mathcal{M}$ denote the set of all possible mutations found in set of sequenced genes $\mathcal{G}$ and $\mathcal{A}$ denote the set of possible alterations in $\mathcal{G}$. Each mutation $m_l \in \mathcal{M}$ can be separated out into a gene component $g_l \in \mathcal{G}$ and alteration $a_l \in \mathcal{A}$. Let $\mathcal{D}$ denote the set of chemotherapy drugs. Two related, albeit different datasets are available to perform the DRP task - labelled pre-clinical cell line data and clinical patient data. Cell line genomic data $X_c \subset \mathcal{P}(\mathcal{M})$ and labelled patient genomic data $X_p \subset \mathcal{P}(\mathcal{M})$, where $\mathcal{P}(.)$ denotes the power set of $\mathcal{M}$. Let $\mathcal{N}_c = |X_c|$ and $\mathcal{N}_p = |X_p|$ denote the number of unique mutation profiles in each dataset. $y_{p(jk)} \in \{0, 1\}$ is a binary RECIST response associated with patient-drug pair $(x_{pj}, d_k)$, while $y_{c(jk)} \in [0, 1]$ is the real-valued AUDRC response for cell line-drug pair $(x_{cj}, d_k)$. To illustrate, a patient mutation profile $x_{p(1)} = \{m_5 = (g_2, a_{10}), m_7 = (g_{100}, a_8)\}$ has a response $y_{p(13)} = 1$ for drug $d_3$. The goal is to predict the response $y_{p(jk)}$ for a new patient-drug pair $(x_{pj}, d_k)$. To achieve this, we perform patient data augmentation, i.e. generate $(X_{aug}, d_{aug}, y_{aug})$ using $(X_c, d_c, y_c)$ and $(X_p, d_p, y_p)$. $d_c$ and $d_p$ denote the set of drugs available in labelled cell line and patient datasets, and $d_{aug} \subseteq \mathcal{D}$. In general, $|d_c| > |d_p|$, $d_c \subseteq \mathcal{D}$ and $d_p \subset \mathcal{D}$, as obtaining drug responses in cell lines for a wide range of drugs is easier than in patients. The real and generated labelled patient data $(X_{aug}, d_{aug}, y_{aug}) \bigcup (X_p, d_p, y_p)$ can then be used to train a downstream DRP classifier $f$. Please note that $*$ can denote $c$ or $p$ in subsequent sections, to denote cell lines and patients respectively.

### 3.2 METHOD OVERVIEW

We propose a *Generative AtteNtion based Data Augmentation and predictive modeLing Framework* - GANDALF, to tackle the labelled patient data scarcity issue via data augmentation. The complete algorithm is available in Algorithm 1. GANDALF generates new patient-like samples from cell lines and assigns them labels in 5 steps - (1) pretraining diffusion models to learn representations of $X_c$ and $X_p$, (2) generating new patient-like samples $X_{aug}$ from $X_c$, (3) training a multi-task learning network using $(X_c, d_c, y_c)$ and $(X_p, d_p, y_p)$, (4) assigning pseudolabels $y_{aug}$ for $(X_{aug}, d_{aug}) \forall d_{aug} \in \mathcal{D}$ and selection of confident samples $(X_s, d_s, y_s) \subseteq (X_{aug}, d_{aug}, y_{aug})$ and (5) training DRP classifier $f$ on $(X_s \cup \mathcal{X}_p', d_s \cup d_p, y_s \cup y_p)$.

The goal is to learn $g(.) : X_{aug} = g(X_c) \sim P(X_p)$, which accounts for patient-specific characteristics. The intuition behind the transformation process is: if we decompose each domain into domain-invariant $Z_s$ and domain-specific $Z_p$ (for patients) and $Z_c$ (for cell lines) representations (Lee & Pavlovic, 2021), to transform $X_c \rightarrow X_p$, we introduce $Z_p$ over $Z_s$ obtained from $X_c$. We can then augment $(X_p, d_p, y_p)$ using $(X_{aug}, d_{aug}, y_{aug}), d_{aug} \in \mathcal{D}$, where $y_{aug}$ can be generated by pseudolabelling (Lee et al., 2013; Kage et al., 2024). Our pseudolabelling approach assumes that $y_c$ and $y_p$ share certain characteristics, while differing in others.

#### 3.2.1 STEP 1: PRETRAINING DIFFUSION MODELS

In this step, we learn $Z_s$, $Z_p$ and $Z_c$ representations. We assume $Z_s \sim \mathcal{N}(0, I)$, which can be modelled using denoising diffusion probabilistic model (DDPM) encoders (Ho et al., 2020). The DDPM decoders learn to remove the domain-specific noise, to reconstruct $X$. Transforming $X_c \rightarrow X_p$ would then involve the use of the patient DDPM decoder on $Z_s$. We train two DDPM models ($TD_p$ and $TD_c$), one per domain, such that they share a common $Z_s$. In addition, we use the pretrained transformer encoder ($T_e$) from (Jayagopal et al., 2024), with padding, to model varying number of mutations. We use domain alignment losses (Sun et al., 2016) to align $Z_s$ and KL-divergence loss to ensure $X_{aug} \sim P(X_p)$. We use cross-attention to ensure $X_{aug}$ retains information from $X_c$.

---

**Algorithm 1** GANDALF training

---

**Require:** Mutation profiles $X_c$, $X_p$, drugs $\mathcal{D}$, cell line-drug labels $y_c$, patient-drug labels $y_p$, time steps t, pre-trained transformer encoder $T_e$, DDPM networks $TD_*$, VAEs $V_*$, pre-train epochs $e_p$, pseudolabel generation epochs $e_s$, upper and lower thresholds $t_u$ and $t_l$ and DRP training epochs $e_d$.

1: **Step 1: Pretraining diffusion models**
2: Obtain transformer embedded samples $Z_{t*} = T_e(X_*) \in \mathcal{R}^{N_* \times k}$
3: Pre-train domain specific VAEs using Eq. 1 and 2
4: **for** $e$ in range($e_p$) **do**
5:     Extract output from the tranformer-VAE encoder network $E = V_{*(e)}(T_e(.))$
    $Z_{v*} = S(\mu_*, \sigma_*)$ $(S(.) = \mu_* + \sigma_* \epsilon$, where $\epsilon \sim \mathcal{N}(0,1), \mu_*, \sigma_* = E_*(X_*))$
6:     $Z_{v*t} = TD_{*(e)}(Z_{v*})$
7:     $\mathcal{X}'_* = denoise(Z_{v*t}, t, TD_{*(d)}(Z_{v*t}))$
8:     $\bar{Z}_{t*} = V_{*(d)}(\mathcal{X}'_*)$
9:     $Z_{Att} = softmax(\frac{Z_{vpt}Z_{vct}^T}{\sqrt{l}})Z_{vct}$
10:     $\hat{Z_{vpa}} = denoise(Z_{Att}, t, TD_{p(d)}(Z_{Att}))$ using Eq. 5
11:     Minimise loss $L_{PRE}$ until convergence.
12: **end for**
13: **Step 2: Generating new patient-like samples**
    $Z_{vct} = TD_{c(e)}(V_{c(e)}(Z_{tc}))$
    $X_{aug} = denoise(Z_{vct}, t, \epsilon_{p\theta}); \epsilon_{p\theta} = TD_{p(d)}(Z_{vct})$
14: **Step 3: Training multi-task learning network**
15: **for** $e$ in range($e_s$) **do**
16:     Obtain cell line and patient embeddings $Z_{v*} = S(E_*(X_*))$
17:     Obtain drug embeddings $Z_{d*} = g_d(d_*)$
18:     For each sample, drug pair concatenate the embeddings to get $O_{*d} = Z_{v*}||Z_{d*}$
19:     Obtain AUDRC and RECIST predictions: $\hat{y}_c = g_a(O_{cd}); \hat{y}_p = g_r(O_{pd})$
20:     Minimise $L_{MTL}$ till convergence.
21: **end for**
22: **Step 4: Assigning pseudolabels and selection of confident samples**
23: $y_{aug} = g_r(X_{aug}||g_d(d_{aug}))$ for $d_{aug} \in \mathcal{D}$.
24: Set $y_{bin}$ as 1 if $y_{aug} >= t_u$, 0 if $y_{aug} < t_l$ and -1 otherwise.
25: Select confident tuples (non-abstained tuples) $(X_s, d_s, y_s)$, i.e. where $y_{bin} \neq -1$.
26: Combine $(X_s, d_s, y_s)$ with $(\mathcal{X}'_p, d_p, y_p)$ to form $(X_{comb}, d_{comb}, y_{comb})$
27: **Step 5: Training drug response prediction classifier**
28: **for** $e$ in range($e_d$) **do**
29:     $\hat{y_{comb}} = f(X_{comb}||d_{comb})$
30:     Minimise loss $L_{BCE}$ in Eq. 10 till convergence.
31: **end for**

---

$T_e$ takes as input $\{m_l; m_l \in \mathcal{M}\}$. Each $m_l$ has two parts - the gene part $g_l \in \mathcal{G}$ and the alteration part $a_l \in \mathcal{A}$. $g_l$ and $a_l$ are tokenized separately, padded and concatenated to generate a per-sample vector. In the embedding step, each $a_l$ is embedded following the variant annotation procedure in (Jayagopal et al., 2024), to obtain a 23-dimensional embedding. This consists of a 17 dimensional binary vector from Annovar (Wang et al., 2010), a 3-dimensional binary vector each from GPD (Li et al., 2020) and ClinVar (Landrum et al., 2018). The embedding for each $a_l$ is passed through a linear layer and concatenated with the corresponding $g_l$ embedding (obtained by one hot encoding), before being fed into $T_e$. The resulting output is mean-aggregated to obtain sample embedding $Z_{t*} = T_e(X_*) \in \mathcal{R}^{N_* \times k}$, where $k$ denotes the maximum sequence length. $k$ is set based on maximum number of alterations in the training data, and all sequences are padded to match $k$. $T_e$ was trained to predict the progression-free survival (PFS) for $(X_p, d_p)$. PFS is indicative of the time after treatment that a cancer patient survives without the cancer progressing. For further details, please refer to (Jayagopal et al., 2024).

To ease training (Rombach et al., 2022), we reduce the dimensionality of $Z_{t*}$ from $k \rightarrow l, l < k$ using variational autoencoders (VAEs) (Kingma & Welling, 2013). We use 2 VAEs - $V_c$ and $V_p$ for cell line and patient domains respectively. These VAEs take as input $Z_{t*} \in \mathcal{R}^{N_* \times k}$ and estimate the mean $\mu_c, \mu_p \in \mathcal{R}^{N_* \times l}$ and standard deviation $\sigma_c, \sigma_p \in \mathcal{R}^{N_* \times l}$ of each domain. Samples generated using the estimated $\mu$ and $\sigma$ are used to train $TD_*$. The VAEs are pretrained on each domain, to minimise reconstruction mean square error and KL divergence loss as in Eq. 1 and 2. The VAE

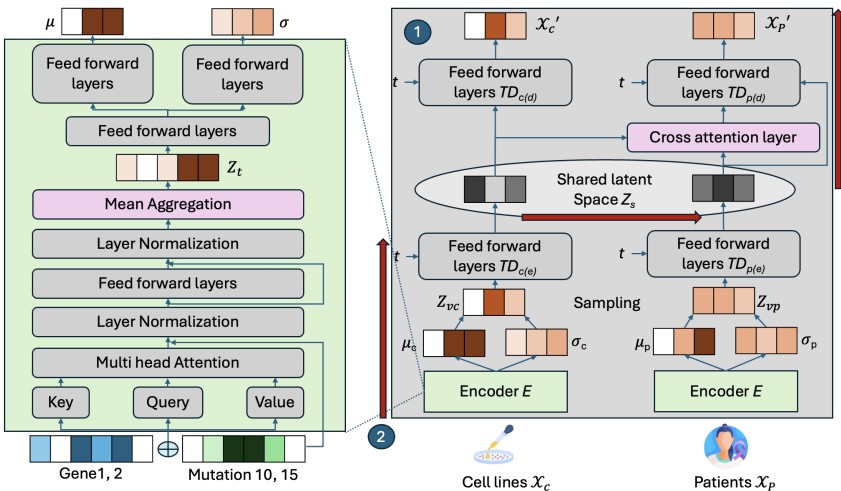

Figure 2: GANDALF architecture used for pretraining domain-specific diffusion models and to generate new patient-like samples using available cell line data. Circled numbers in blue indicate steps from Algorithm 1.

pretraining loss is $L_{VAE} = L_R + L_{KLD}$.

$$L_R = \frac{1}{N_*}\Sigma_{N_*}(\hat{Z}_{t*} - Z_{t*})^2 \tag{1}$$

$$L_{KLD} = -(0.5/N_*)\Sigma_{N_*}(1 + log(\sigma_*(Z_{t*})^2) - \mu_*(Z_{t*})^2 - \sigma_*(Z_{t*})^2) \tag{2}$$

where $N_*$ denotes number of mutation profiles ($\mathcal{N}_c$ or $\mathcal{N}_p$), $\hat{Z}_{t*}$ is the reconstructed VAE output. Pretrained $T_e$ attached to the encoder layers of the pretrained $V_c$ and $V_p$, are henceforth referred to as encoder networks $E_c$ and $E_p$; $\mu_*, \sigma_* = E_*(X_*)$. Parameters of $T_e$ are frozen for training.

The sampled output from $E_*$, $Z_{v*} = S(\mu_*, \sigma_*)$ $\big(S(.) = \mu_* + \sigma_*\epsilon$ denotes VAE sampling, where $\epsilon \sim \mathcal{N}(0, I)\big)$ is fed into $TD_c$ and $TD_p$, with encoder $TD_{*(e)}$ and decoder $TD_{*(d)}$. Since $Z_{v*}$ is a vector, we used feed forward linear layers in $TD_*$ (Kotelnikov et al., 2023). To learn $Z_s$, we perform domain alignment, using CORAL loss (Sun et al., 2016). CORAL loss minimises the co-variance between the latent spaces, as in Eq. 4. Although in theory, DDPM encoders should yield isotropic Gaussians as $T \to \infty$, the use of CORAL loss enforces that the two domains share $Z_s$, when $T$ is finite. $TD_c$ and $TD_p$ are trained jointly with the CORAL loss using $L_{ALIGN} = L_{DDPM} + L_{CORAL}$, as in Eq. 3 and 4.

$$L_{DDPM} = E_{(Z_{vc}, \epsilon_c, t)}[\epsilon_c - \epsilon_{c\theta}(Z_{vct}, t)]^2 + E_{(Z_{vp}, \epsilon_p, t)}[\epsilon_p - \epsilon_{p\theta}(Z_{vpt}, t)]^2 \tag{3}$$

$$L_{CORAL} = \Sigma_l\Sigma_l||C(Z_{vct}) - C(Z_{vpt})||^2; C(Z) = \frac{1}{n}\Sigma_n(Z_i - \bar{Z}_i)(Z_i - \bar{Z}_i)^T \tag{4}$$

$\epsilon_c$ and $\epsilon_p$ are ground truth noise distributions added to $X_c$ and $X_p$. $Z_{vct} = TD_{c(e)}(Z_{vc})$ and $Z_{vpt} = TD_{p(e)}(Z_{vp})$ are the noisy representations after $t$ timesteps through $TD_{*(e)}$. $\epsilon_{c\theta}$ and $\epsilon_{p\theta}$ are estimated by $TD_{*(d)}$. $\bar{Z}$ denotes mean. $Z_{v*t}$ is denoised using $\epsilon_{*\theta}$ (Eq. 5) to obtain $\mathcal{X}'_c$ and $\mathcal{X}'_p$. These are passed through VAE decoders to obtain $\bar{Z}_{tc} = V_{c(d)}(\mathcal{X}'_c)$ and $\bar{Z}_{tp} = V_{p(d)}(\mathcal{X}'_p)$. $\beta_t$ in Eq. 5 is the variance schedule (Nichol & Dhariwal, 2021) of $\epsilon_c$ and $\epsilon_p$ at diffusion step time $t$.

$$\mathcal{X}'_c = denoise(Z_{vct}, t, \epsilon_{c\theta}); \mathcal{X}'_p = denoise(Z_{vpt}, t, \epsilon_{p\theta})$$
$$where\ denoise(X_t, t, \epsilon) = \frac{1}{\sqrt{\hat{\alpha}_t}}(X_t - \sqrt{1 - \hat{\alpha}_t}\epsilon); \hat{\alpha}_t = \Pi_{i=1}^t(\alpha_i); \alpha_t = 1 - \beta_t \tag{5}$$

To ensure that $X_{aug}$ preserves information from $X_c$, we use cross-attention Rombach et al. (2022). Given, $Z_{vct}$ and $Z_{vpt}$, we obtain $Z_{Att} = softmax(\frac{Z_{vpt}Z_{vct}^T}{\sqrt{l}})Z_{vct}$. $Z_{Att}$ pays attention to $Z_{vct}$. $Z_{Att}$ is passed through $TD_{p(d)}$ and denoised using $\epsilon_{p\theta}$ to obtain $\hat{Z_{vpa}}$. A KL divergence loss $L_{KLDA}$ is also calculated between the distributions of $Z_{vp}$ and $\hat{Z_{vpa}}$ to ensure eventual adherence to $P(X_p)$, as

in Equation 6. Additional mean square error terms $L_{MSE}$ between $Z_{t*}$ and $\bar{Z}_{t*}$ and KL divergence terms $L_{KLDV}$ for $Z_{v*}$ are calculated as in Equation 7.

$$L_{KLDA} = 0.5\Sigma_{N*}(-1 + log(\sigma(\hat{Z_{vpa}}^2)) - log(\sigma(Z_{vp})^2) + exp(log(\sigma(Z_{vp})^2))$$
$$-log(\sigma(\hat{Z_{vpa}}^2))) + (\mu(Z_{vp}) - \mu(\hat{Z_{vpa}}))^2 exp(-log(\sigma(\hat{Z_{vpa}}^2)))) \tag{6}$$

$$L_{MSE} = \frac{1}{N_*}\Sigma_{N_*}(Z_{t*} - \bar{Z}_{t*})^2$$
$$L_{KLDV} = -(0.5/N_*)\Sigma_{N_*}(1 + log(\sigma_*(Z_{v*})^2) - \mu_*(Z_{v*})^2 - \sigma_*(Z_{v*})^2) \tag{7}$$

The overall training loss is $L_{PRE} = L_{ALIGN} + L_{KLDA} + L_{KLDV} + L_{MSE}$. Architecture details are available in Figure 2. The training is done in an unsupervised manner and does not require labeled data.

### 3.2.2   STEP 2: GENERATING NEW PATIENT-LIKE SAMPLES

To generate $X_{aug}$, we run inference on the trained model using $X_c$. $X_c$ is first passed through $T_e$, followed by $V_{c(e)}$, to get $Z_{vc}$. This is then passed through $TD_{c(e)}$ to get $Z_{vct}$. This is analogous to removing $Z_c$ from the input samples. As the latent spaces of the DDPMs are already aligned, $Z_{vct}$ can be denoised using $TD_{p(d)}$ to obtain $X_{aug}$. This step corresponds to introducing $Z_p$ to $Z_s$. The red arrows in Figure 2, indicates the generation of $X_{aug}$ from $X_c$.

### 3.2.3   STEP 3: TRAINING MULTI-TASK LEARNING NETWORK

In this step, the goal is to train a network to assign $y_{aug}\forall(X_{aug}, d_{aug}), d_{aug} \in \mathcal{D}$. A naive approach would involve training a classifier $\hat{f}$ on $(X_p, d_p, y_p)$ and using it to predict $y_{aug}$. However, $d_p \subset d_{aug}$, since only a small subset of drugs are provided to patients as per clinical guidelines. This implies that $P(X_p, d_p, y_p)$ learnt by $\hat{f}$ may not fully model $P(X_p \cup X_{aug}, d_p \cup d_{aug}, y_p \cup y_{aug})$. During inference, $\hat{f}$ may encounter drugs outside of the training set, yielding noisy $y_{aug}$. A similar constraint exists in using weak supervision methods (Ratner et al., 2017; Zhang et al., 2022) to assign pseudo-labels. Further, $\hat{f}$ can be prone to overfitting, given the small size of $(X_p, d_p, y_p)$.

In this step we alleviate overfitting concerns using larger data $(X_c, d_c, y_c)$, in a multi-task learning (MTL) setup, with additional regularizing loss terms. Moreover, $d_c \simeq \mathcal{D}$, which allows the network to learn from drugs $\notin d_p$. We also capture the shared traits between $y_c$ and $y_p$ by projecting labelled $(X_c, d_c)$ and $(X_p, d_p)$ into a shared latent space $O_s$, and capture the differences, via two separate prediction heads - a classification head $\hat{y}_p = g_r(X_p, d_p) \in \{0,1\}$ and a regression head $\hat{y}_c = g_a(X_c, d_c) \in [0,1]$. $O_s$ is learnt by aligning the latent representations, using CORAL loss (Sun et al., 2016), as in Equation 8. $X_c$ and $X_p$ are first passed through the pretrained encoder network $E(.)$ to obtain $\mu_c$, $\mu_p$, $\sigma_c$ and $\sigma_p$. Sampling $S$ is applied as before to obtain $Z_{vc}$ and $Z_{vp}$. $Z_{vc}$ and $Z_{vp}$ are concatenated with drug embeddings obtained from a feedforward multi-layer perceptron (MLP) $Z_{d*} = g_d(d_*) \in \mathcal{R}^{\mathbf{N}_* \times l}$. The resulting concatenated representations $O_{cd} = Z_{vc}||Z_{dc} \in \mathcal{R}^{\mathbf{N}_c \times 2l}$ and $O_{pd} = Z_{vp}||Z_{dp} \in \mathcal{R}^{\mathbf{N}_p \times 2l}$ where $||$ denotes concatenation, $\mathbf{N}_c = |(X_c, d_c, y_c)|$ and $\mathbf{N}_p = |(X_p, d_p, y_p)|$ denote number of labelled sample, drug pairs ($\mathbf{N}_p < \mathbf{N}_c$).

$$L_{CORAL\_O} = \Sigma_{2l}\Sigma_{2l}||C(O_{cd}) - C(O_{pd})||^2; \quad C(Z) = \frac{1}{n}\Sigma_n(Z_i - \bar{Z}_i)(Z_i - \bar{Z}_i)^T \tag{8}$$

$O_{cd}$ is passed through a feed-forward MLP $g_a$ to predict AUDRC values $\hat{y}_c = g_a(O_{cd})$. $O_{pd}$ is passed through another feed forward MLP $g_r$ to predict RECIST values $\hat{y}_p = g_r(O_{pd})$. The entire network is trained to minimise $L_{MTL} = L_{BCE} + L_{MSE} + L_{CORAL\_O}$ as in Equation 9, where $\sigma(x) = \frac{1}{1+e^{-x}}$. MTL architecture is shown in Figure 3(left).

$$L_{BCE} = -\frac{1}{\mathbf{N}_p}\Sigma_{\mathbf{N}_p}[y_p log(\sigma(\hat{y}_p)) + (1 - y_p)log(1 - \sigma(\hat{y}_p))]; \quad L_{MSE} = \frac{1}{\mathbf{N}_c}\Sigma_{\mathbf{N}_c}(y_c - \hat{y}_c)^2 \tag{9}$$

### 3.2.4   STEP 4: ASSIGNING PSEUDOLABELS AND SELECTION OF CONFIDENT SAMPLES

To obtain $y_{aug}$, we first generate all possible $\mathcal{N}_c \times |\mathcal{D}|$ pairs $(X_{aug}, d_{aug}), d_{aug} \in \mathcal{D}$. We pass the drug representation $d_{aug}$ through $g_d$. We concatenate the resulting drug embedding $g_d(d_{aug})$ with $X_{aug}$. This is then passed through $g_r$ and $\sigma(.)$ to get $y_{aug} \in [0,1]$, as shown in Figure 3(right, top).

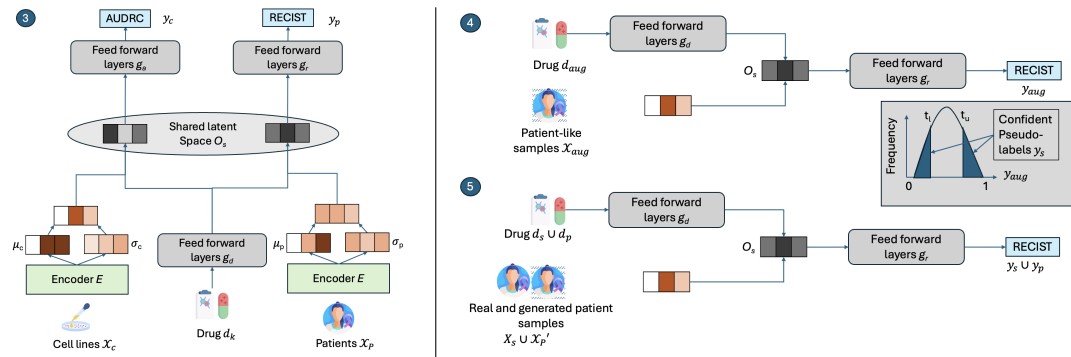

Figure 3: GANDALF architecture for multi-task training (left), pseudolabel generation and selection of confident samples (right, top) and training downstream DRP model (right, bottom). Circled numbers in blue indicate steps from Algorithm 1.

$(X_{aug}, d_{aug}, y_{aug})$ may however be noisy due to incorrect predictions from $g_r$. Prior work on subset selection (Lang et al., 2022) has identified that choosing a subset of more confident pseudolabelled samples is more effective than using the complete pseudolabelled dataset. We use $y_{aug}$, to select this subset. $y_{bin}$ is generated by binning $y_{aug}$ into 3 groups, using an upper and lower threshold $t_u$ and $t_l$. $y_{bin} = 1$, if $y_{aug} >= t_u$; $y_{bin} = 0$, if $y_{aug} < t_l$ and $y_{bin} = -1$ otherwise (abstained samples). Only $\mathbf{N}_s < (\mathcal{N}_c \times |\mathcal{D}|)$ high confidence (non-abstained) samples ($y_{bin} \neq -1$) are used for the downstream DRP classifier training.

### 3.2.5 STEP 5: TRAINING DRUG RESPONSE PREDICTION CLASSIFIER

The non-abstained, high confidence generated "patient"-drug pairs after pseudo labeling $((X_s, d_s, y_s)$ of size $\mathbf{N}_s)$ are combined with $\mathbf{N}_p$ $(\mathcal{X'}_p, d_p, y_p)$ pairs to train a drug response predicting feed forward neural network $f$ (Figure 3, right, bottom). $f$ is trained to minimise BCE loss in Eq. 10.

$$L_{BCE} = -\frac{1}{\mathbf{N}_p + \mathbf{N}_s} \Sigma_{\mathbf{N}_p + \mathbf{N}_s} [y_i log(\sigma(\hat{y_i})) + (1 - y_i) log(1 - \sigma(\hat{y_i}))] \quad (10)$$

GANDALF offers several advantages. The use of VAEs and DDPMs makes the model generative in nature. While generation in DDPMs usually involves sampling from $\mathcal{N}(0, I)$ and denoising, here the sampling incorporates prior knowledge from $X_c$. This also enables the use of $(X_c, d_c, y_c)$ in generating pseudo-labels for $X_{aug}$. When $\mathbf{N_s} > 0$, it reduces chances of overfitting.

## 4 EXPERIMENTS AND RESULTS

### 4.1 DATASETS

We used publicly available cell line and patient datasets, for all our experiments. Cell line mutation profiles were obtained from the Cancer Cell Line Encyclopedia (CCLE) DepMap (v23Q4) (Ghandi et al., 2019; Barretina et al., 2012). AUDRC responses were obtained from the GDSCv2 (Iorio et al., 2016; Yang et al., 2012). Patient mutation profiles and associated response labels for drugs were collected from The Cancer Genome Atlas (TCGA) (Weinstein et al., 2013), CbioPortal (CBIO) (Harding et al., 2019; Nixon et al., 2019; de Bruijn et al., 2023; Gao et al., 2013; Cerami et al., 2012) and UC SanDiego Moores Cancer Center (Moores) (Schwaederle et al., 2016). Patient response, measured via RECIST were coalesced into binary labels (1: positive response; 0: negative) (Peres da Silva et al., 2021). Drugs were encoded using 2048 dimensional binary Morgan fingerprints (Morgan, 1965). We exclude samples on multiple drug regimen and retain only patients given a single drug at a time. This results in 1197 CCLE samples, 541 TCGA, 44 Moores and 84 CBIO patient samples with documented response labels for 211 drugs in cell lines and 56 drugs across patients. We restrict our analysis to the 324 genes found in a popular clinical sequencing panel, FoundationOne CDx (Milbury et al., 2022) and removed samples without mutations in these genes. We also removed samples with responses to drugs without a Morgan fingerprint. For the transformer pretraining, we used 71 non-small cell lung cancer and 71 colorectal cancer samples from GENIE (Choudhury et al., 2023; Garcia et al., 2023), with a documented progression-free survival. We had a total of 156441 train, 17371 validation and 21589 test cell line, drug pairs. We also had 488/488/487 train, 53/54/56 validation and 115/114/113 test patient, drug pairs over 3 folds (folds 0/1/2 respectively) (details in Appendix Section A.1).

Table 1: Performance comparison across SOTA drug response prediction methods. Best performing results are highlighted in bold, while the second best performing results are underlined.

| AUROC (Mean ± Standard deviation) | | | | | |
|---|---|---|---|---|---|
| Method | Cis | Flu | Gem | Pac | Tem |
| GANDALF | $\underline{0.6343 \pm 0.0306}$ | $\mathbf{0.7309 \pm 0.0664}$ | $\mathbf{0.6188 \pm 0.0674}$ | $\mathbf{0.7728 \pm 0.1253}$ | $\mathbf{0.6451 \pm 0.0776}$ |
| DruID | $\mathbf{0.6764 \pm 0.1447}$ | $0.6071 \pm 0.1988$ | $\underline{0.5092 \pm 0.1005}$ | $0.5119 \pm 0.2324$ | $0.6194 \pm 0.0420$ |
| PANCDR | $0.6278 \pm 0.0308$ | $0.4762 \pm 0.1798$ | $0.4429 \pm 0.2268$ | $0.4236 \pm 0.4168$ | $\underline{0.6436 \pm 0.2310}$ |
| PREDICT-AI | $0.5072 \pm 0.0331$ | $0.3869 \pm 0.0372$ | $0.5046 \pm 0.1181$ | $\underline{0.6815 \pm 0.1786}$ | $0.5350 \pm 0.0606$ |
| drug2tme | $0.5243 \pm 0.1301$ | $\underline{0.7167 \pm 0.1957}$ | $0.4568 \pm 0.0857$ | $0.3194 \pm 0.3127$ | $0.5951 \pm 0.2541$ |
| WISER | $0.4622 \pm 0.1685$ | $0.6095 \pm 0.193$ | $0.4305 \pm 0.0867$ | $0.3641 \pm 0.2522$ | $0.5297 \pm 0.0738$ |
| CODE-AE | $0.6322 \pm 0.1872$ | $0.5381 \pm 0.1606$ | $0.5085 \pm 0.0503$ | $0.3611 \pm 0.3155$ | $0.4332 \pm 0.3123$ |
| AUPRC (Mean ± Standard deviation) | | | | | |
| Method | Cis | Flu | Gem | Pac | Tem |
| GANDALF | $\underline{0.9093 \pm 0.0355}$ | $\mathbf{0.8483 \pm 0.0933}$ | $\mathbf{0.5874 \pm 0.175}$ | $\mathbf{0.9558 \pm 0.024}$ | $0.2535 \pm 0.1108$ |
| DruID | $\mathbf{0.9176 \pm 0.0671}$ | $0.7588 \pm 0.1484$ | $0.4515 \pm 0.1297$ | $\underline{0.8897 \pm 0.0223}$ | $0.3014 \pm 0.1039$ |
| PANCDR | $0.9018 \pm 0.0324$ | $0.6951 \pm 0.1530$ | $0.4562 \pm 0.2270$ | $0.8561 \pm 0.1019$ | $\underline{0.3049 \pm 0.2653}$ |
| PREDICT-AI | $0.8622 \pm 0.0189$ | $0.5885 \pm 0.0581$ | $0.3873 \pm 0.0489$ | $0.8687 \pm 0.1090$ | $0.1373 \pm 0.0050$ |
| drug2tme | $0.8754 \pm 0.0523$ | $\underline{0.8092 \pm 0.1722}$ | $\underline{0.4826 \pm 0.0947}$ | $0.7824 \pm 0.1023$ | $\mathbf{0.3058 \pm 0.1327}$ |
| WISER | $0.8454 \pm 0.0685$ | $0.7505 \pm 0.0657$ | $0.3901 \pm 0.0885$ | $0.7724 \pm 0.1585$ | $0.1762 \pm 0.0243$ |
| CODE-AE | $0.9059 \pm 0.0521$ | $0.6665 \pm 0.1435$ | $0.4735 \pm 0.0701$ | $0.8208 \pm 0.0574$ | $0.1756 \pm 0.0929$ |

## 4.2 Comparison with cancer drug response prediction methods

We compared GANDALF against 4 recent state-of-the-art (SOTA) methods which take sample, drug pairs as model inputs, namely, DruID (Jayagopal et al., 2023), PREDICT-AI (Jayagopal et al., 2024), drug2tme (Zhai & Liu, 2024) and PANCDR (Kim et al., 2024). We also compared GANDALF against CODE-AE (He et al., 2022) and WISER (Shubham et al., 2024), which train separate models per drug. We report performance metrics on 5 drugs, with samples available in all 3 test folds, namely Cisplatin (Cis), Paclitaxel (Pac), 5-Fluorouracil (Flu), Gemcitabine (Gem) and Temozolomide (Tem). We do drug-specific model tuning in GANDALF, by only augmenting with sample, drug pairs for the drug considered. For CODE-AE and WISER, we train separate models per drug. Apart from GANDALF, only PREDICT-AI could handle varying length inputs. For all other methods, we converted the mutation profiles into fixed length input vectors of 7776 dimensions, following the pre-processing in (Jayagopal et al., 2023). Validation set correlation between predicted and actual response was used for early stopping and hyper-parameter selection. As shown in Table 1, GANDALF achieves the best AUROC in Flu, Gem, Pac and Tem and second-best in Cis. GANDALF achieves the best AUPRC score in Flu, Gem and Pac, and second-best in Cis.

## 4.3 Ablation study

Next, we performed an ablation study to empirically verify the importance of each component in the architecture. We successively removed each component and measured the overall AUROC and AUPRC performance across all the drugs in the test set. The key components of GANDALF are the MTL network for pseudolabeling, cross-attention in pretraining DDPMs and use of transformers to model varying length inputs. We first removed the cell line head in the MTL network (*W/O MTL*). Next, we removed the cross-attention KL divergence loss $L_{KLDA}$ (*W/O cross-attention*). We then removed the use of pretrained transfomer (*W/O transformer*) in the input to the network and instead used the 7776 dimensional input used by other SOTA methods. The full model with all components shows the best performance in terms of both AUROC and AUPRC, highlighting the importance of each component in the overall performance (Table 2, Ablation). The above ablation removes each component successively from the architecture. In Appendix A.5, we have also included ablation tests where only one component is removed at a time. We also analyse test performance sensitivity to increased volume of pseudolabelled data; details in Appendix Section A.2. A low to moderate volume of high confidence samples is better than large volume of low confidence samples.

## 4.4 Comparison with other augmentation strategies

There are no known label-invariant mutation data augmentation approaches for cancer DRP (refer Section 2.2 for details). As a baseline, we compare GANDALF against a naive data augmentation approach (Lee et al., 2023), where we perturb the 7776 dimensional inputs, using samples from $\mathcal{N}(0, I)$. This is done once per patient, drug pair (*W perturbation*) in the training data, and the associated label is assumed to remain the same as in the original sample, resulting in a dataset of size $2\mathbf{N}_p$. In addition, we also compare GANDALF against a vanilla feed-forward MLP (*W/O*

Table 2: Contribution of various components (ablation) in GANDALF, comparisons with other augmentation and pseudolabeling strategies.

| Experiment | Method | AUROC (mean $\pm$ std) | AUPRC (mean $\pm$ std) |
|---|---|---|---|
| Ablation | GANDALF | **0.8409 $\pm$ 0.0437** | **0.778 $\pm$ 0.0255** |
| | *W/O MTL* | 0.753 $\pm$ 0.1637 | 0.6448 $\pm$ 0.1604 |
| | *W/O cross-attention* | 0.752 $\pm$ 0.165 | 0.6443 $\pm$ 0.1636 |
| | *W/O transformer* | 0.6007 $\pm$ 0.08 | 0.5632 $\pm$ 0.1101 |
| Augmentation | *W perturbation* | 0.6306 $\pm$ 0.0255 | 0.5967 $\pm$ 0.0611 |
| | *W/O aug* | 0.6052 $\pm$ 0.0219 | 0.5784 $\pm$ 0.0394 |
| Pseudolabeling | *W majority vote* | 0.8153 $\pm$ 0.0541 | 0.756 $\pm$ 0.0827 |

*aug*), trained using only $(X_p, d_p, y_p)$. We compare the learning curves (Appendix Figure 6) and test performance metrics (Table 2, Augmentation). In both cases, we fix training epochs. In all folds, no augmentation and Gaussian perturbation strategies result in overfitting, where the validation loss show an increase while the training loss remains low. This is consistent with the fact that smaller datasets can result in overfitting. The test performance metrics for these methods is lower than that of GANDALF. The slight improvement due to perturbation indicates the benefit of data augmentation in improving overall performance.

## 4.5 COMPARISON WITH MAJORITY VOTE BASED PSEUDOLABELING

We compared MTL based pseudolabeling strategy against another pseudolabeling strategy similar to Dong-DongChen & WeiGao (2018). The augmented data $(X_{aug}, d_{aug}, y_{aug})$ is passed through 3 separate feed-forward networks, trained on $(X_p, d_p, y_p)$. The pseudolabels generated by each network is aggregated by majority voting (Lang et al., 2022). As before, non-abstained samples are used to train the downstream DRP model, along with $(X_p, d_p, y_p)$. The results comparing GANDALF against this approach (*W majority vote*) are shown in Table 2, Pseudolabeling. While the majority voting strategy does perform well, GANDALF outperforms it in overall AUROC and AUPRC. This may be potentially due to the use of the larger cell line labelled data, with more drugs, as opposed to the smaller labelled patient dataset.

## 5 CONCLUSIONS AND DISCUSSION

In this paper, we propose GANDALF, a generative patient data augmentation framework, to tackle the challenge of training a cancer DRP model with limited labelled data.Unlike prior DRP methods that augment data in the shared space between patients and cell lines, we utilise the larger labelled cell line dataset to generate more patient-like samples as well as their pseudo-labels. GANDALF outperforms SOTA DRP methods, and also shows improved performance when compared to baseline genomic data augmentation and pseudo labeling approaches. GANDALF has a large number of parameters and sub-modules, each of which needs pretraining, increasing overall training time. Learning the underlying data distributions is limited by available labelled cell lines and patients.

There are several future directions to explore, which may improve GANDALF further. In this paper, we have only considered labelled patient profiles for training, although the pretraining stage supports unlabelled data. Future work can evaluate the use of unlabelled patient profiles in all steps of training. We examined the quality of the generated samples by comparing the distributions against the original patient data. More extensive studies to examine the biological significance of the generated samples and their fidelity can shed light on the patterns captured by the model. Generative strategies, which can incorporate known biological information on co-occurring mutations, can also be explored in the future. In the cell line datasets we used, we have included both solid and non-solid tumor types, that can lead to differences in pharmacological responses (Basu et al., 2013; Yao et al., 2018; Gerdes et al., 2021; Sharifi-Noghabi et al., 2021b). The effect of these tumor types on model performance can be examined in the future. We could even build tumor type specific models by fine-tuning the existing model using data specific to each cancer type. Overall, GANDALF sets the stage for using generative techniques in the field of cancer DRP research, and emphasises the importance of capturing patient domain-specific characteristics for improving downstream prediction performance.

## 6 REPRODUCIBILITY

Our code and data are made publicly available at `https://github.com/ajayago/GANDALF`.

ACKNOWLEDGMENTS

This research is supported by the National Research Foundation, Singapore under its AI Singapore Programme (Award Number: AISG-100E-2023-116). Anand D. Jeyasekharan is supported by the Ministry of Health, Singapore, through the NMRC Clinician Scientist Award (MOH-CSAINV20nov-0010). Aishwarya Jayagopal is supported by the National University of Singapore Research Scholarship. We would like to acknowledge the American Association for Cancer Research in the development of the AACR Project GENIE registry, as well as members of the consortium for their commitment to data sharing. Images in this paper were created using FlatIcon and Freepik.

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

# A APPENDIX

## A.1 EXPERIMENT SETTINGS

### A.1.1 DRUG SELECTION CRITERIA

The patient dataset we used had 56 drugs. For each of the 56 drugs in patients, we first consider those with at least 20 labelled patient samples (He et al., 2022) - this reduced labelled data to 12 drugs. For each drug, we divided the samples into groups based on cancer type and data source. Each group with $> 20$ samples was divided into 2:1 ratio in 3-fold label based stratified cross-validation. For some groups, no test samples were available. We excluded these to get 7 drugs. These drugs were used in the ablation studies in Table 2, to report overall performance metrics. We removed drugs which had $< 3$ positive samples as it would cause issues in CV, where one fold may have test samples with only a single label - this resulted in the five drugs shown in Table 1.

### A.1.2 TRAIN-TEST SPLIT

RECIST labels in patients were initially coalesced into 2 groups - Complete and Partial Response as label 1 (good response), Stable and Progressive Disease as label 0 (bad response). The labelled patient samples obtained were grouped based on the drug, cancer type and source of dataset (TCGA, Moores, CBIO). Each group with $\geq 20$ samples was divided into 3-fold cross validation train-test splits, stratified by label. Groups with $< 20$ samples were only used for training. The train and test labelled samples across all groups and folds were combined to form 3 train-test folds respectively. Each of the 3 train folds were further divided in a 90:10 ratio to obtain a train-validation split. Cell line data was also grouped in a similar fashion and divided into a single train-validation and test fold. The training and evaluation in all cases use sample, drug pairs where the sample could be from either domain. We had a total of 156441 train, 17371 validation and 21589 test cell line, drug pairs. We also had 488/488/487 train, 53/54/56 validation and 115/114/113 test patient, drug pairs over the 3 folds. We run inference on test patient, drug pairs, and report the average AUROC and AUPRC metrics across 3 test folds.

## A.2 SENSITIVITY TO VOLUME OF PSEUDOLABELLED DATA

We examined the sensitivity of the overall model performance to increasing the quantity of pseudolabelled data. We change the amount of pseudolabelled data by varying the upper and lower thresholds $t_u$ and $t_l$. Increasing $t_l$ and decreasing $t_u$ is equivalent to adding more pseudolabelled samples. We varied $t_l$ from 0.1 to 0.4, $t_u$ from 0.5 to 0.9, in increments of 0.1. In all cases, only a single parameter was changed while all others were left constant. Figure 4 indicates that a lower value of $t_l$ shows better performance. This may result in fewer non-abstained samples with label 0, and improve confidence in the samples selected for the downstream DRP task. A higher $t_u$ in general improves performance with 0.8 yielding the best. If $t_u$ is too low or $t_l$ is too high, it may admit more low-confidence samples, $y_{aug}$ being closer to 0.5. If $t_u$ is too high, it may drastically reduce the number of positive labels available for downstream DRP training, also reducing performance. Table 3 indicates the number of pseudolabelled samples added in each case.

## A.3 SENSITIVITY TO DIFFERENT AMOUNTS OF TRAINING DATA

We conducted two experiments to evaluate the effect of varying amounts of real and synthetic patient data.

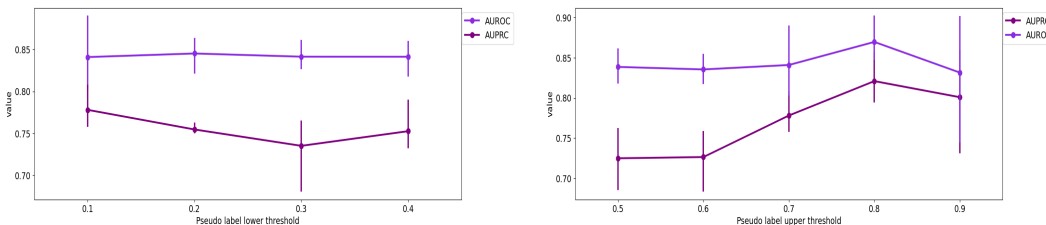

Figure 4: Sensitivity tests on value of pseudo label lower (left) and upper (right) thresholds.

| Lower threshold values | Fold 0 pseudolabelled responders / non-responders | Fold 1 pseudolabelled responders / non-responders | Fold 2 pseudolabelled responders / non-responders |
|---|---|---|---|
| 0.1 | 3830/60101 | 874/15668 | 241/7157 |
| 0.2 | 3830/192454 | 874/125098 | 241/81011 |
| 0.3 | 3830/355849 | 874/323572 | 241/274803 |
| 0.4 | 3830/481589 | 874/479348 | 241/462177 |
| Upper threshold values | Fold 0 pseudolabelled responders / non-responders | Fold 1 pseudolabelled responders / non-responders | Fold 2 pseudolabelled responders / non-responders |
| 0.5 | 29599/60101 | 25932/15668 | 25554/7157 |
| 0.6 | 9568/60101 | 6336/15668 | 4023/7157 |
| 0.7 | 3830/60101 | 874/15668 | 241/7157 |
| 0.8 | 1578/60101 | 27/15668 | 0/7157 |
| 0.9 | 500/60101 | 0/15668 | 0/7157 |

Table 3: Number of pseudolabelled samples used in sensitivity test of thresholds.

### A.3.1  EFFECT OF VARYING AMOUNTS OF PSEUDOLABELLED DATA

We retain all the real train patient data and randomly sample 25%, 50%, 75% and 100% of the generated, confident pseudolabelled data, and use this in training the DRP model. 0% setting indicates no augmented data in the DRP training. Results are shown in the Table 4. 0% does the worst, without any augmentation. Best AUROC is at 50% addition of pseudolabelled data, best AUPRC at 25% pseudolabelled data. Across 25-100% settings, the difference in performance is not statistically significant. For the case of 0% vs any other level of augmentation, differences are statistically significant, indicating that adding pseudolabelled data improves performance. To answer the question of how much pseudolabelled is helpful we will need further studies on possibly larger datasets.

### A.3.2  EFFECT OF VARYING AMOUNTS OF REAL PATIENT DATA

We randomly sample 25%, 50%, 75% and 100% of real labelled patient data. In each case we sample twice the number of real samples from the pseudolabelled data. 100% setting thus refers to

| % of pseudolabelled data | Average AUROC over 3 folds | Average AUPRC over 3 folds | Number of pseudolabelled patient data (fold 0) | Number of real labelled patient data (fold 0) |
|---|---|---|---|---|
| 0% | $0.5263 \pm 0.0195$ | $0.5229 \pm 0.0249$ | 0 | 488 |
| 25% | $0.8584 \pm 0.0361$ | $0.7838 \pm 0.0564$ | 15983 | 488 |
| 50% | $0.8613 \pm 0.0279$ | $0.7796 \pm 0.0437$ | 31966 | 488 |
| 75% | $0.8577 \pm 0.0269$ | $0.7677 \pm 0.0354$ | 47948 | 488 |
| 100% | $0.8409 \pm 0.0437$ | $0.778 \pm 0.0255$ | 63931 | 488 |

Table 4: Performance comparison for varying quantities of pseudolabelled data

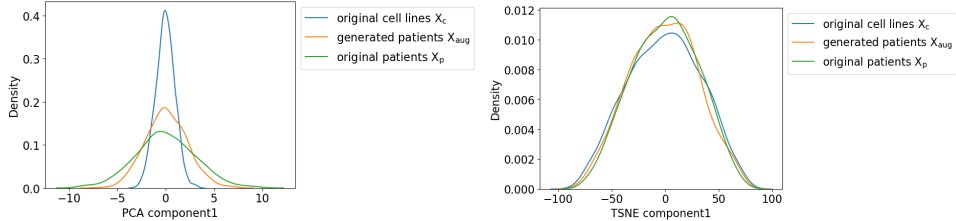

Figure 5: Kernel Density Estimation plot comparing the distribution of first PCA component (left) and first TSNE component (right) across original cell line, real patient and generated patient data

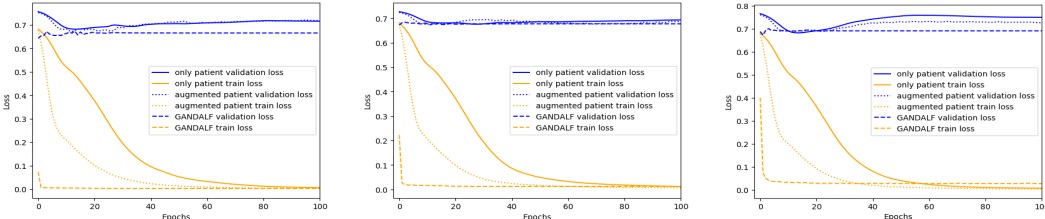

Figure 6: Learning curves on (left to right) 3 cross-validation folds, with orange line indicating train loss and blue indicating validation loss. Dotted lines indicate augmentation with Gaussian perturbation, solid lines indicate no augmentation, dashed lines indicate GANDALF augmentation.

3 times the size of real labelled patient data. In general, as seen in Table 5, as more real labelled data is added performance improves, as generally expected.

## A.4 HYPERPARAMETER SELECTION

For baseline models, we used the hyperparameter ranges defined in each of the papers. We did a hyperparameter sweep over these ranges using Bayesian Optimization for maximum of 15 runs, to determine the best hyperparameters for our dataset. We did not tune epochs since we had early stopping in all cases. Across methods, we focused on the last stage of DRP for tuning. For DruID, MTL learning rate range was [0.001, 0.05], RECIST prediction network dimensions for 1st hidden layer were tuned in 64, 32 and dimensions for second hidden layer in 16, 8. In PREDICT-AI, we tuned learning rate in the range [0.0001, 0.001], batch size in 128, 64. In PANCDR, we tuned encoder bottleneck dimensions in 100, 128, 256, GCN dimensions in 100, 128, 256, learning rate in [0.0001, 0.001], adversarial learning rate in [0.0001, 0.001], lambda in 1, 0.1, 0.01, batch size in 128, 256. In CODE-AE and WISER, we tuned dropout in 0, 1. In WISER we additionally tuned learning rate in the range [0.001, 0.1].

For GANDALF, we mainly focused on the hyperparameters in the supervised training stages, key being the lower and upper thresholds and learning rate parameters for the DRP and MTL models. We varied the lower threshold between 0.1 to 0.5 and upper threshold from 0.5 to 0.9, with increments done based on quantiles calculated from predicted probability of response after MTL training. This was done for each drug separately. The hidden layers from the VAE were set to 64 dimensions

| % of real data (pseudolabelled data = 2 x real data) | Average AUROC over 3 folds | Average AUPRC over 3 folds | Number of pseudo labelled patient data (fold 0) | Number of real labelled patient data (fold 0) |
|---|---|---|---|---|
| 25% | $0.5326 \pm 0.0152$ | $0.5239 \pm 0.0222$ | 244 | 122 |
| 50% | $0.581 \pm 0.0216$ | $0.5505 \pm 0.0274$ | 488 | 244 |
| 75% | $0.6888 \pm 0.0257$ | $0.638 \pm 0.0348$ | 732 | 366 |
| 100% | $0.7086 \pm 0.0247$ | $0.6533 \pm 0.0374$ | 976 | 488 |

Table 5: Performance comparison for varying quantities of real patient data

Table 6: Performance comparison across different ablation tests, where each test removes one component from GANDALF. Best performing results are highlighted in bold.

| AUROC (Mean ± Standard deviation) | | | | | |
|---|---|---|---|---|---|
| Method | Cis | Flu | Gem | Pac | Tem |
| GANDALF | **0.6343 ± 0.0306** | **0.7309 ± 0.0664** | **0.6188 ± 0.0674** | **0.7728 ± 0.1253** | 0.6451 ± 0.0776 |
| *W/O MTL* | 0.3409 ± 0.219 | 0.5333 ± 0.075 | 0.5587 ± 0.1787 | 0.2758 ± 0.1461 | **0.7513 ± 0.0805** |
| *W/O cross-attention* | 0.6061 ± 0.0475 | 0.7309 ± 0.0834 | 0.6188 ± 0.0674 | 0.7728 ± 0.1253 | 0.6152 ± 0.1074 |
| *W/O transformer* | 0.3735 ± 0.1404 | 0.4143 ± 0.1122 | 0.5718 ± 0.0805 | 0.5625 ± 0.3903 | 0.2106 ± 0.0457 |
| *W/O VAE* | Out of memory issues | | | | |
| *W/O DDPM* | 0.4849 ± 0.0909 | 0.6929 ± 0.1189 | 0.5162 ± 0.1247 | 0.5208 ± 0.4161 | 0.3138 ± 0.0647 |
| *W/O pseudolabels* | 0.6048 ± 0.1185 | 0.6452 ± 0.2304 | 0.6019 ± 0.1891 | 0.6825 ± 0.3345 | 0.5026 ± 0.1647 |
| AUPRC (Mean ± Standard deviation) | | | | | |
| Method | Cis | Flu | Gem | Pac | Tem |
| GANDALF | 0.9093 ± 0.0355 | **0.8483 ± 0.0933** | **0.5874 ± 0.175** | **0.9558 ± 0.024** | 0.2535 ± 0.1108 |
| *W/O MTL* | 0.8101 ± 0.0793 | 0.7345 ± 0.1 | 0.5697 ± 0.0628 | 0.7582 ± 0.1012 | **0.3215 ± 0.1623** |
| *W/O cross-attention* | **0.9183 ± 0.0255** | 0.8483 ± 0.0967 | 0.5873 ± 0.1753 | 0.9558 ± 0.024 | 0.2068 ± 0.0585 |
| *W/O transformer* | 0.8047 ± 0.0417 | 0.6400 ± 0.1231 | 0.4760 ± 0.0865 | 0.8478 ± 0.1374 | 0.0993 ± 0.0195 |
| *W/O VAE* | Out of memory issues | | | | |
| *W/O DDPM* | 0.8696 ± 0.0098 | 0.8241 ± 0.0741 | 0.4932 ± 0.1867 | 0.8273 ± 0.1636 | 0.1150 ± 0.0263 |
| *W/O pseudolabels* | 0.919 ± 0.035 | 0.8146 ± 0.1368 | 0.5669 ± 0.1321 | 0.9066 ± 0.1301 | 0.2702 ± 0.1045 |

based on our GPU memory restrictions and batch size was 512. For the transformer we used 64 dimensional embeddings, 4 heads and 8 encoder layers. For the cell line VAE, we used 1024, 128 and 64 hidden units and for patient VAE we used 512, 128 and 64 hidden units. Both VAEs used tanh activation. The DDPM uses linear layers with dimensions as the VAE representation size, and uses dropout and ReLU. $\beta_t$ was set based on the cosine beta scheduling in (Nichol & Dhariwal, 2021). The MTL network uses 2 linear layers each in embedding drugs, predicting RECIST and predicting AUDRC, with ReLU activation. Max epochs were set to 500 for pretraining, 100 for the MTL and DRP training. Early stopping was done using patient validation set pearson correlation as in (Sharifi-Noghabi et al., 2021a).

## A.5 Additional Ablation Experiments

To understand the contribution of each individual component, we added additional ablation studies where each test removes just one component of the architecture. We also performed drug specific tuning in each case. Table 6 shows the results of each ablation test. Apart from the ablation tests in Table 2, we also added three more tests *W/O VAE*, *W/O DDPM* and *W/O pseudolabels*. In *W/O VAE*, we attempt to directly feed the output of the transformer encoder layer to the domain-specific DDPMs, bypassing the VAEs. In *W/O DDPM*, we replace the DDPMs with two domain-specific VAEs. The data augmentation is done by passing the cell lines through the cell line VAE encoder and the patient VAE decoder. The pseudolabelling and downstream DRP training remains the same as GANDALF in both cases. In *W/O pseudolabels*, to remove the influence of pseudolabelled data, we directly use the MTL part of the network (after stage 3) to run inference on the test patient data. The use of transformers and MTL appear to contribute the most to the model performance. The use of pseudolabelled data also helps improve average performance in most cases.

## A.6 Comparison of distribution

We examine the distribution of the $X_{aug}$ with respect to the real distributions of $\mathcal{X}'_c$ and $\mathcal{X}'_p$. We expect $X_{aug}$ to be closer to the patient distribution than the original cell lines, while also retaining information from the original cell line data. Each dataset is further subjected to principal component analysis (principal components (PCs) from $\mathcal{X}'_p$) to obtain lower dimensional representations for easier visualization. Figure 7 (top, right) shows that original cell line data had lower variance compared to the real patient data (Figure 7, top left). However, the generated patient data (Figure 7, top middle) is closer to the real patient data in terms of the variance of data points. This indicates that the generated data captures patient-specific heterogeneity. A similar trend is seen in the density plots of

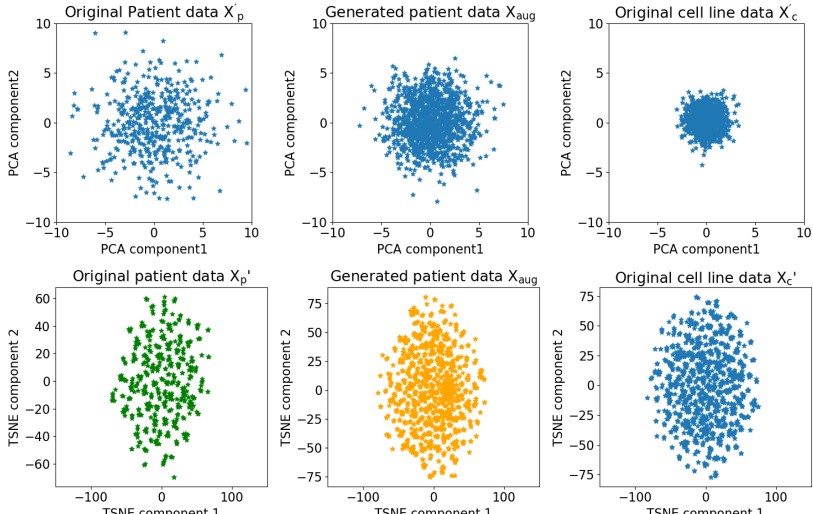

Figure 7: Comparison of distribution (left to right) across real and generated data, using PCA (top) and TSNE (bottom) methods.

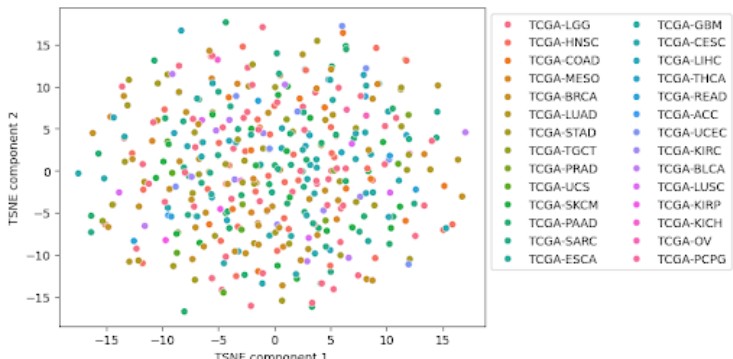

Figure 8: TSNE plots of first two components of the patient data in the representation space, color coded based on TCGA cancer types.

the first PC in Appendix Figure 5. Quantitatively, we also examine the Kolmogorov–Smirnov (KS) test between the PCs of the 3 distributions. KS distance statistic between generated patient data and real patient data over 3 folds is $0.0694 \pm 0.0071$, while the same between original cell line data and patients is $0.2524 \pm 0.0022$. The PCs of the augmented data is closer to that of the real patient data, when compared to the distance between the PCs of the original cell line and patient data. This indicates that the augmented data starts resembling the patient data while retaining information from the original cell line data.

## A.7 CHECKING FOR BATCH EFFECTS IN THE REPRESENTATION SPACE

Our patient data comes from three different sources - TCGA, CBIO and Moore's. To ensure that these representations do not inadvertently capture batch effects, we perform a TSNE based visualization, where the patient latent representations are colored based on the cancer type (as coded in TCGA). For Moore's and CBIO datasets, we identified the corresponding category in TCGA. Figure 8 shows the TSNE plot for the first two components, after embedding the patient data into the representation space. The lack of well defined boundaries across cancer types (indicated by various colors) suggest that there is no batch effect across the mutation datasets.

| Cancer type | AUROC over 3 folds | AUPRC over 3 folds |
|---|---|---|
| TCGA-BRCA | $0.8947 \pm 0.0368$ | $0.8720 \pm 0.0712$ |
| TCGA-CESC | $0.3197 \pm 0.1403$ | $0.7704 \pm 0.0716$ |
| TCGA-HNSC | $0.7652 \pm 0.137$ | $0.9788 \pm 0.0137$ |
| TCGA-STAD | $0.7119 \pm 0.1318$ | $0.8253 \pm 0.1222$ |
| TCGA-PAAD | $0.6620 \pm 0.0765$ | $0.6239 \pm 0.0516$ |
| TCGA-LGG | $0.4309 \pm 0.0563$ | $0.1406 \pm 0.0061$ |

Table 7: Comparison of performance across various cancer types.

## A.8 PERFORMANCE ACROSS CANCER TYPES

During the train-test split, we split the data based on cancer type and drug. Then we divided each group into 2:1 ratio if more than 20 samples were present per group. The train data thus contained all available cancer types. The evaluation was on a limited set of cancer types - 'TCGA-BRCA', 'TCGA-CESC', 'TCGA-HNSC', 'TCGA-STAD', 'TCGA-PAAD', 'TCGA-LGG'. Performance per cancer type from existing test splits - we calculated the metrics over the available test splits by grouping based on cancer type. Table 7 shows the results.

