# OpenReview forum: "GANDALF: Generative AttentioN based Data Augmentation and predictive modeLing Framework for personalized cancer treatment"
_ICLR.cc/2025/Conference — ICLR 2025 Poster_

### Official Review · Reviewer_NotN · 2024-10-20

**Soundness:** 4
**Presentation:** 3
**Contribution:** 3
**Rating:** 8
**Confidence:** 4

**Summary:**

The study shows an AI4Science application. It majorly focuses on using related cell line data and patients' clinical data to do drug and dosage response. The major contributions of this paper are: 1. proposing a composite framework (diffusion $+$ transformer) architecture to use cell line data to enrich limited patient clinical data. 2. the proposed framework also include a prediction module for the drug response prediction task and it beats SOTA models on the task.

**Strengths:**

- The study introduces a complete framework to do the DRP task. The framework uses a smart way to use the cell-line data to augment the patient data in order to improve model's performance on the target task.
- I like the idea that projects the patient embedding and cell-line data embedding data on the shared latent space and uses the cross attention strategy to keep the shared information.
- The step 3 is also a smart move that addresss the limited drug issue in patients' clinical dataset.
- I can see from Table 1 that the proposed framework achieves general improvement on the datasets compared with baselines.
- The authors provided comprehensive ablation studies. Table 2 demonstrates the importance of each component of the framework.

**Weaknesses:**

- In figure 4, the aim is to show distribution similarity between generated patient data. But the PCA visualization is a linear projection method, and I cannot tell any difference between the distributions. Can you use some non-linear methods UMAP/t-SNE to show the distributions, and mark the labels using different colors? It would be better to see the distribution characteristics and differences.
- Why you put related works at last? It is hard for a reader not in this field to follow the work. It would be better put the section right after the introduction. And unsupervised models has been introduced well in the paper, can you elaborate a little bit more on the brief definitions or examples of inductive and transductive approaches in the context of drug response prediction, and how their method differs from yours.
- One key step is data augmentation. The author does provide the sample size for train/test/val in the appendix, but I didn't find the details about the sample size, i.e. how many psedo-patient data were added before training the final model. I think it is a key information for your experiments. And, I like your experiments in the appendix A.2, but the quantity there confused me. It would be better to provide the sample size directly rather than the thresholds.

**Questions:**

As I am not an expert in this application, I want to left the decision to the other reviewers and chairs that whether it deserves a highlight. But I think it is a very interesting application of diffusion models to this two modalities.
Good work!

---

> ### Author Response · Authors · 2024-11-24
>
> Thank you for your comments and suggestions. We have incorporated your suggestions in the paper. Please let us know if additional clarifications are needed.
>
> **W.1: TSNE plots**
>
> We have used TSNE plots apart from PCA for Figure 4. The KDE plots of the first TSNE component show that generated patients lie close to the original patient distribution. We have added the plots (Figure 5 and 7 in revised paper) to the paper.
>
> **W.2: Related work**
> * We have moved related work to the top (Section 2 in revised paper). Thank you for the suggestion.
> * Inductive methods use labelled data from both cell line and patient domains. Methods like AITL, drug2tme, TCRP use this approach. Here they may either use multi-task learning approaches or few shot learning to capture the differences in label distribution across the two domains.
> * Transductive methods like TUGDA, WISER and PANCDR use labelled data from cell lines and unlabelled patient data. Here the assumption is that the response label does not change across the domains. To this end, most papers convert the continuous valued cell line response to discrete categories as seen in patients, using arbitrary thresholds.
> * GANDALF is an inductive method using multi task learning to model the distribution differences in labels across the two domains.
>
> **W.3: Sample sizes for sensitivity test**
>
> Thank you for raising this. We have included the number of pseudo labelled data that has been added in each of the thresholds in the table below (Table 3 in revised paper).
>
> |Lower threshold value| Fold 0 psuedolabelled responders/non-responders|Fold 1 psuedolabelled responders/non-responders|Fold 2 psuedolabelled responders/non-responders|
> |-|-|-|-|
> |0.1|3830/60101|874/15668|241/7157|
> |0.2|3830/192454|874/125098|241/81011|
> |0.3|3830/355849|874/323572|241/274803|
> |0.4|3830/481589|874/479348|241/462177|
>
> |Upper threshold value| Fold 0 psuedolabelled responders/non-responders|Fold 1 psuedolabelled responders/non-responders|Fold 2 psuedolabelled responders/non-responders|
> |-|-|-|-|
> |0.5|29599/60101|25932/15668|25554/7157|
> |0.6|9568/60101|6336/15668|4023/7157|
> |0.7|3830/60101|874/15668|241/7157|
> |0.8|1578/60101|27/15668|0/7157|
> |0.9|500/60101|0/15668|0/7157|

---

> > ### Comment · Reviewer_NotN · 2024-11-24
> > **Reply to Official Comments**
> >
> > Thank you for your response. I have no more questions. I have slightly increased my confidence score.

---

> > > ### Author Response · Authors · 2024-11-24
> > >
> > > Thank you very much for your review and feedback!

---

### Official Review · Reviewer_1XQK · 2024-10-26

**Soundness:** 2
**Presentation:** 2
**Contribution:** 3
**Rating:** 6
**Confidence:** 4

**Summary:**

Authors proposed GANDALF a method for data augmentation that takes labeled cell line and patients mutation data as input and makes drug response prediction via generating patient-driven samples in a 5 step process:
- pretraining diffusion models on cell lines and patients
- generating augmented samples
- training a multitask predictor using labeled cell lines and patients
- making predictions for augmented samples and selecting confident ones
- training a final predictor on confident augmented samples and patients

GANDALF was compared with state-of-the-art of drug response prediction across 5 chemotherapy drugs in terms of AUROC and AUPR and was able to outperform the majority of them.

**Strengths:**

- The introduction of biological aspects was gentle and informative enough for those not familiar with the motivating application of the work.
- While the separate components of GANDALF are not novel per se, components are linked together in a novel way for data augmentation - based on patient data for precision oncology.
- The experimental results show improvements compared to baselines across multiple drugs

**Weaknesses:**

- The input patient data is itself coming from 3 different resources i.e, P(Xp1) != P(Xp2)!=P(Xp3) similarly, the cell line data is also coming from two resources (input from CCLE, output from GDSC) but there is no indication of that in neither the assumptions nor the setup or the method. What makes GANDALF capable of handling shifts like that?

- Although authors have provided the source code, critical information is missing with respect to reproducibility of this work. For example, it is unclear if the reported numbers are from how many independent runs or hyper-parameters and the procedure for tuning them are not reported in the paper for any of the components e.g., DDPA, transformers, VAE, MTL, etc.

- Citations of the utilized data are inaccurate and incomplete. For example, for CCLE and GDSC the seminal work like
“Barretina et al., The Cancer Cell Line Encyclopedia enables predictive modelling of anticancer drug sensitivity 2012” or “Iorio et al. A Landscape of Pharmacogenomic Interactions in Cancer 2016” are not cited. Moreover, Although authors claimed that they’ve used GDSCv2, the citation is for 2012 which is more correlated with GDSCv1 timeline. Is this a citation error or GDSCv1 was used?

- Authors used genomic data from CCLE and response data from GDSC, there is no justification for why the response profiles of these datasets are interchangeable. How many cell lines were in common between CCLE and GDSC, was the drug screening assay comparable between the two? this is important because GDSCv1 used Syto60 assay while CCLE and GDSCv2 used CellTiter-Glo (going back to the previous point on which GDSC was used).

- There are parts of the paper that is challenging to follow for example, “We had a total of 156441 train, 17371 validation and 21589 test cell line, drug pairs. We also had 488/488/487 train, 53/54/56 validation and 115/114/113 test patient, drug pairs over the 3 folds.” Is unclear to me. Is this a reference to different patient data sources?

- W/o MTL in ablation seems misleading as it only removes the cell line part but the proper way of doing this is to replace it with a drug specific single task model train on both cell lines and patients.

- Another missing ablation is w/o diffusion what happens if you apply VAE directly to the input data?

- GANDALF learning curve is missing in Figure 7

**Questions:**

- There is no information regarding the applicability domain of GANDAL including what types of cancers was this method train/evaluated on? What was the performance per cancer type? similarly for cell lines, was the data from only solid tumours or both solid and non-solid tumours?

- While somatic mutations can be predictive of drug response, other available data types such as RNA-Seq make more accurate predictions. To my knowledge, CCLE/GDSCv2 and TCGA/CbioPortal also have gene expression data, if the ultimate goal is to assist clinicians with choice of therapy why did you choose not to use other data types?

- Why is GANDALF only applicable to chemotherapy drugs? Mutation data is also predictive of targeted therapeutics I can imagine it could be due to lack of labeled data for targeted drugs if so, isn’t this a huge disadvantage for the proposed method that requires labeled patient data as input? For targeted drugs like Erlotinib (Byers et al. An Epithelial-Mesenchymal Transition Gene Signature Predicts Resistance to EGFR and PI3K Inhibitors and Identifies Axl as a Therapeutic Target for Overcoming EGFR Inhibitor Resistance 2013) there are many labeled cell lines and smaller labeled patient data. Do you have any sense on the minimum required number of cell lines and patient samples for your method? I believe an experiment of percentage of labeled samples would be highly beneficial.

- The reported performance for WISER and Code-AE in this work are substantially lower than those reported in the original papers for similar drugs and datasets. It seems WISER and Code-AE had a similar setup so what’s the difference between your setup and their’s?

I'd be happy to revise my score if authors answer all questions/concerns thoroughly particularly:
- potential data issue
- applicability domain
- reproducibility
- missing ablations

---

> ### Author Response · Authors · 2024-11-24
>
> Thank you for your in-depth feedback and comments. Please find our responses below. We hope we have addressed all your concerns. Please let us know if any additional clarification is needed from us.
>
> **WEAKNESSES**
>
> **W.1: Datasets used**
> * For cell lines, we use the mutation profiles from CCLE DepMap portal, with responses from GDSCv2[1, 3], similar to [2]. This involves matching cell line names across both datasets and is established in prior literature[3].
> * For clinical samples, we are using only the somatic mutation data, which is a binary data that has been "normalized/subtracted" from their normal..
> * GANDALF uses latent representations of patients and cell lines for the downstream modeling. To ensure that these representations do not inadvertently capture batch effects, we perform a TSNE based visualization, where the patient latent representations are colored based on the cancer type (as coded in TCGA). For Moore’s and CBIO datasets, we identified the corresponding category in TCGA. The lack of well defined boundaries across types suggest that there is no batch effect across the mutation datasets. We have included this in the Appendix A.7 (Figure 8) of the revised paper.
>
> **W 2: Hyperparameters**
>
> Thank you for raising this. The hyperparameter details are available in our code repo, and hence we had not included them in the earlier draft. We have added this to the appendix section A.4 in the revised paper. Details of hyperparameters:
> * For baseline models, we used the hyperparameters defined in each of the papers.
> * For GANDALF, we mainly focused on the hyperparameters in the supervised training stages, key being the lower and upper thresholds and learning rate parameters for the DRP and MTL models.
> * We varied the lower threshold between 0.1 to 0.5 and upper threshold from 0.5 to 0.9, with increments done based on quantiles calculated from predicted probability of response after MTL training. This was done for each drug separately.
> * The hidden layers from the VAE were set to 64 dimensions based on our GPU memory restrictions and batch size was 512.
> * For the transformer we used 64 dimensional embeddings, 4 heads and 8 encoder layers.
> * For the cell line VAE, we used 1024, 128 and 64 hidden units and for patient VAE we used 512, 128 and 64 hidden units. Both VAEs used tanh activation.
> * The DDPM uses linear layers with dimensions as the VAE representation size, and uses dropout and ReLU.
> * The MTL network uses 2 linear layers each in embedding drugs, predicting RECIST and predicting AUDRC, with ReLU activation.
> * We did a hyperparameter sweep using Bayesian Optimization for a maximum of 15 runs, using WandB.
> * Max epochs were set to 500 for pretraining, 100 for the MTL and DRP training. Early stopping was done using patient validation set pearson correlation as in [4].
>
> **W.3: GDSC version clarification**
>
> We have checked the references and updated these in section 4.1. Thank you for pointing this out. We have used GDSCv2 for our analysis, and this was a citation error. Apologies for this.
>
> **W.4: Cell line datasets**
>
> For cell lines, we use the mutation profiles from CCLE DepMap portal, with responses from GDSCv2[1], similar to [2]. This involves matching cell line names across both datasets and is established in prior literature[3].
>
> **References**
>
> [1] Yang, W., Soares, J., Greninger, P., Edelman, E. J., Lightfoot, H., Forbes, S., ... & Garnett, M. J. (2012). Genomics of Drug Sensitivity in Cancer (GDSC): a resource for therapeutic biomarker discovery in cancer cells. Nucleic acids research, 41(D1), D955-D961.
>
> [2] He, D., Liu, Q., Wu, Y., & Xie, L. (2022). A context-aware deconfounding autoencoder for robust prediction of personalized clinical drug response from cell-line compound screening. Nature Machine Intelligence, 4(10), 879-892.
>
> [3] Iorio, F., Knijnenburg, T. A., Vis, D. J., Bignell, G. R., Menden, M. P., Schubert, M., ... & Garnett, M. J. (2016). A landscape of pharmacogenomic interactions in cancer. Cell, 166(3), 740-754.
>
> [4] Sharifi-Noghabi, H., Harjandi, P. A., Zolotareva, O., Collins, C. C., & Ester, M. (2021). Out-of-distribution generalization from labelled and unlabelled gene expression data for drug response prediction. Nature Machine Intelligence, 3(11), 962-972.

---

> ### Author Response · Authors · 2024-11-24
>
> **W.5: Train-test data splits**
> * We have clarified this in the revised paper, section 4.1. The numbers correspond to the number of sample, drug pairs (either patient or cell line) found in the train, validation, test splits. The 3 CV splits of patients had 488 patient, drug pairs in training fold 0, 488 pairs in training fold 1 and 487 pairs in training fold 2, 53, 54 and 56 pairs in validation fold 0, 1 and 2; and 115, 114 and 113 pairs in test fold 0, 1, and 2 respectively. The train, val, test split for cell line, drug pairs had 156441 train, 17371 validation and 21589 test cell line, drug pairs.
> * After processing the data as detailed in Appendix A.1, we had cell line, drug pairs and patient, drug pairs, which were then used for training and validating the model.
> * To this end, we performed a stratified 3-fold cross validation split of the patient, drug labeled data into train-test folds. Further, each train fold was divided into a train-validation split as well. The cell line, drug pairs were divided into train, validation and test split but only one such split was created since our focus is mainly on patients and how the model performs on patient data.
>
> **W.6: Clarifications on MTL**
> * We would like to clarify that the multi-task setup in GANDALF refers to the two drug response tasks, i.e. predicting AUDRC in cell lines and RECIST in patients.
> * Here, tasks refer to these two related tasks, albeit of different domains and data distributions.
> * We would like to emphasise that tasks here do not correspond to different drugs.
> * As such, in the ablation W/O MTL, we sought to convert it to a single task setting, by removing the AUDRC prediction task head from the architecture (as in Table 2).
> * Please let us know if you need further details on this.
>
> **W.7: Ablation without DDPM**
> * Thank you for raising this.
> * We have now conducted an ablation study removing DDPMs (Table 6 in revised paper). Our experiments show that DDPMs are important to the overall architecture.
> * We used VAEs instead of DDPMs. We have two domain specific VAEs, and the augmentation is done by passing the latent space representation of cell lines (obtained from the cell line VAE encoder) through the patient VAE decoder.
> * This augmented data is then pseudolabelled as before, using the MTL setup and downstream DRP training is done as in GANDALF. The results, as in the table below, show empirically that the use of DDPMs help the model.
>
> |Model|Cis||Flu||Gem||Pac||Tem||
> |-|-|-|-|-|-|-|-|-|-|-|
> ||AUROC|AUPRC|AUROC|AUPRC|AUROC|AUPRC|AUROC|AUPRC|AUROC|AUPRC|
> |GANDALF with tuning|0.6343 $\pm$ 0.0306|0.9093 $\pm$ 0.0355|0.7309 $\pm$ 0.0664|0.8483 $\pm$ 0.0933|0.6188 $\pm$ 0.0674|0.5874 $\pm$ 0.175|0.7728 $\pm$ 0.1253|0.9558 $\pm$ 0.024|0.6451 $\pm$ 0.0776|0.2535 $\pm$ 0.1108|
> |GANDALF W/O DDPM|0.4849 $\pm$ 0.0909|0.8696 $\pm$ 0.0098|0.6929 $\pm$ 0.1189|0.8241 $\pm$ 0.0741|0.5162 $\pm$ 0.1247|0.4932 $\pm$ 0.1867|0.5208 $\pm$ 0.4161|0.8273 $\pm$ 0.1636|0.3138 $\pm$ 0.0647|0.1150 $\pm$ 0.0263|
>
> **W.8: Learning curves**
>
> We will update the plots on Figure 7 (Figure 6 in revised paper) with GANDALF added in.
>
> **QUESTIONS**
>
> **Q.1: Performance across cancer types**
> * We split the data based on cancer type and drug. Then we divided each group into ⅔-⅓ ratio if more than 20 samples were present per group. The train data thus contained all available cancer types.
> * The evaluation was on a limited set of cancer types - 'TCGA-BRCA', 'TCGA-CESC', 'TCGA-HNSC', 'TCGA-STAD', 'TCGA-PAAD', 'TCGA-LGG'
> * Performance per cancer type from existing test splits - we calculated the metrics over the available test splits by grouping based on cancer type are shown below (Table 7 in revised paper)
>
>
> | Cancer type | AUROC over 3 folds | AUPRC over 3 folds |
> |--|--|--|
> |TCGA-BRCA|0.8947 $\pm$ 0.0368|0.8720 $\pm$ 0.0712|
> |TCGA-CESC|0.3197 $\pm$ 0.1403|0.7704 $\pm$ 0.0716|
> |TCGA-HNSC|0.7652 $\pm$ 0.137|0.9788 $\pm$ 0.0137|
> |TCGA-STAD|0.7119 $\pm$ 0.1318|0.8253	$\pm$ 0.1222|
> |TCGA-PAAD|0.6620 $\pm$ 0.0765|0.6239 $\pm$ 0.0516|
> |TCGA-LGG|0.4309 $\pm$ 0.0563|0.1406 $\pm$ 0.0061|
>
> We used all available cell line data (solid and non-solid)
>
> **Q.2: Choice of data type**
> * Though more powerful in predicting drug response, RNASeq is not yet widely used in clinical settings.
> * Our focus was on building a predictive model that can use the data available in clinical next generation sequencing panels like FoundationOne CDx, as in [1]. As such we only focused on the 324 genes sequenced in such a panel as FoundationOne CDx and only on mutations that are available in these panels.
>
> **References**
>
> [1] Jayagopal, A., Xue, H., He, Z., Walsh, R. J., Hariprasannan, K. K., Tan, D. S. P., ... & Rajan, V. (2024, August). Personalised Drug Identifier for Cancer Treatment with Transformers using Auxiliary Information. In Proceedings of the 30th ACM SIGKDD Conference on Knowledge Discovery and Data Mining (pp. 5138-5149).

---

> > ### Comment · Reviewer_1XQK · 2024-11-25
> > **Data issues + cancer-specific performance**
> >
> > Thank you for your comprehensive rebuttal and extra experiments. I am still concerned about the way you are using these datasets:
> > - If you are using GDSCv2 response data with CCLE genomic data then logically you are limited to cell lines in common between the two as you mentioned above which pending on resources and/or preprocessing would be ~400 cell lines in common between them however in the submission (line 409) after the preprocessing it says "1197 CCLE samples" which should have been a much smaller number or I am missing something.
> > - Solid and non-solid tumors have significantly different pharmacological behaviour i.e., different response data [1-4]. The impact of this on GANDALF is unknown and needs to be elaborated or at least discussed.
> >
> > - I am also concerned about applicability of the method. New cancer-specific results clearly shows that for two cancer types the method is worse than a random predictor (AUROC). Does this mean GANDALF should be used for breast cancer or head and neck cancer only?
> >
> > [1] Basu et al. An interactive resource to identify cancer genetic and lineage dependencies targeted by small molecules. Cell 2013.
> >
> > [2] Yao F et al. Tissue specificity of in vitro drug sensitivity. J Am Med Inform Assoc 2017.
> >
> > [3] Gerdes H, Casado P, Dokal A, et al. Drug ranking using machine learning systematically predicts the efficacy of anti-cancer drugs. Nature Communications 2021.
> >
> > [4] Sharifi-Noghabi et al. Drug sensitivity prediction from cell line-based pharmacogenomics data: guidelines for developing machine learning models. Briefings in Bioinformatics 2021.

---

> > > ### Author Response · Authors · 2024-11-25
> > >
> > > Thank you for your feedback! Please find our responses below, which we hope will clarify things:
> > >
> > > **Regarding the CCLE samples**: According to DepMap, the mutations profiles are indexed by Profile ID. By following the mapping guide, we mapped the Profile ID to Model ID, to which the cell line metadata corresponds. After combining with GDSCv2 we had 799 cell lines in common. More specifically, 1197 omics profiles correspond to these cell line samples with response.
> > >
> > > **Differences across solid and non-solid tumors**: Thank you for raising this! We will add this in the discussion section of the paper, as something to explore in the future.
> > >
> > > **Cancer type specific performance**: In the numbers reported in this case, we have not done any cancer type specific tuning. As such we cannot concretely say that GANDALF can(not) work for specific types. As you have mentioned, there may be significant differences across tumour types and this would need to be considered while creating cancer type specific models before clinical deployment. This may be another avenue to explore in the future, to build cancer type specific models as well.

---

> > > > ### Comment · Reviewer_1XQK · 2024-11-25
> > > >
> > > > Sorry but I am still confused about number of samples. Your method requires labelled cell lines for training, how did you use the entire 1197 omics data for training?
> > > >
> > > > Regarding the cancer-specific results, I am not clear on how you want to add cancer-specific fine-tuning to your method as there is no indication of that in the submission.
> > > >
> > > > On a separate note (sorry I forgot to mention that in my previous comment), the validation learning curves for GANDALF seem mostly to be flat (for GANDALF loss) and the model overfits fairly quickly.
> > > >
> > > > One suggestion for authors for your future submissions is to post response to questions/comments that are related to clarifications or misunderstandings earlier. With the level of engagement for this paper, I think we could have benefitted more from a longer discussion.
> > > >
> > > > I want to thank authors and other reviewers for their engagement towards the end of the rebuttal period. I think the submission has merits and new points added by the authors helped a lot for understanding this work better. I am more inclined to maintain my current score as data (partially) and the applicability of the method are not fully resolved for me. Nonetheless, rest assured that I will read authors' last remarks due on Nov 27th.

---

> > > > > ### Author Response · Authors · 2024-11-26
> > > > >
> > > > > Thank you for the questions. Please find our responses below:
> > > > >
> > > > > **Cell line dataset**:
> > > > >
> > > > > We used the profile IDs to map to Model IDs, and associated Model IDs with the cell line names, which were then used to check against the GDSCv2 dataset. There were 1197 profiles in the DepMap data we downloaded, which corresponded to 799 cell lines. Each profile ID was mapped to a response from GDSC, resulting in 1197 labelled profiles. These were then used for steps 1-3 of GANDALF training. Please note that the labelled cell lines are not used in the final DRP model training. Steps 4 and 5 only use patient data (both real and synthetic) and do not use cell line data.
> > > > >
> > > > > **Cancer type specific models and loss curves**:
> > > > >
> > > > > We could perform cancer type specific tuning in the last stage of DRP training (step 5), where we can select just the samples from a specific cancer type and finetune the existing model using a few epochs (similar to the training in DruID).
> > > > >
> > > > > In the loss curves, we specifically allowed the models to train for more epochs to examine the loss curves - in the versions of GANDALF in Table 1 and 2,  we perform early stopping to avoid overfitting.
> > > > >
> > > > > We will add this to the discussion of our paper. Thank you for raising this!

---

> > > > > > ### Comment · Reviewer_1XQK · 2024-11-27
> > > > > > **Thank you**
> > > > > >
> > > > > > Thank you for clarifying these points.
> > > > > > I suggest to crop the learning curves to the first 50 (I assume epochs, the figure did not have legend for the x axis) and clarify which loss was exactly used for early stopping if you haven't done already.
> > > > > > Again, I really appreciate your tenacity during the rebuttal phase. I have updated my score accordingly.

---

> > > > > > > ### Author Response · Authors · 2024-11-28
> > > > > > >
> > > > > > > Thank you for your insightful feedback, which has greatly helped us improve our work. We will update the loss curve to include the first 50 epochs, and add a label for the X axis. We have also included the details for early stopping in section 4.2.
> > > > > > >
> > > > > > > We are very grateful for your review, thank you!

---

> ### Author Response · Authors · 2024-11-24
>
> **Q.3: Drug selection and amount of data**
>
> **Why is GANDALF only applicable to chemotherapy drugs?**
>
> We chose the drugs based on the availability of labelled patient data. If sufficient data is available in the future, we could also predict for other drugs. Details of drug selection for evaluation is available in Appendix A.1.1
>
> **Isn’t it a huge disadvantage to require labeled patient data as input?**
>
> Labeled patient data is crucial for cancer DRP models, predicting response in patients. This is due to the difference in treatment response measured in patients and cell lines. While a few previous methods have tried to use only unlabelled patient data, they make assumptions on the distribution of patient response data, based on cell line responses. These responses are however unavailable in a clinical setting.
>
> **How much labeled data is needed?**
>
> To understand the impact of varying quantities of labelled real and synthetic patient data, we conducted two experiments (Appendix A.3 in revised paper).
>
>
> **Experiment A:**
> * We retain all the real train patient data and randomly sample 25%, 50%, 75% and 100% of the generated pseudolabelled data, and use this in training the DRP model.
> * 0% setting indicates no augmented data in the DRP training. Results are shown in the table below.
> * 0% does the worst, without any augmentation. Best AUROC is at 50% addition of pseudolabelled data, best AUPRC at 25% pseudolabelled data.
> * Across 25-100% settings, the difference in performance is not statistically significant. For the case of 0% vs any other level of augmentation, differences are statistically significant, indicating that adding psuedolabelled data improves performance. To answer the question of how much pseudolabelled is helpful we will need further studies on possibly larger datasets.
>
> |% of pseudolabelled data|Average AUROC over 3 folds|Average AUPRC over 3 folds|Number of pseudo labelled patient data (fold 0) | Number of real labelled patient data (fold 0)|
> |-|-|-|-|-|
> |0%|0.5263 $\pm$ 0.0195 | 0.5229 $\pm$ 0.0249 |0|488|
> |25%|0.8584 $\pm$ 0.0361|0.7838 $\pm$ 0.0564|15983|488|
> |50%|0.8613 $\pm$ 0.0279|0.7796 $\pm$ 0.0437|31966|488|
> |75%|0.8577 $\pm$ 0.0269|0.7677 $\pm$ 0.0354|47948|488|
> |100%|0.8409 $\pm$ 0.0437|0.778 $\pm$ 0.0255|63931|488|
>
> **Experiment B:**
> * We randomly sample 25%, 50%, 75% and 100% of real labelled patient data. In each case we sample twice the number of real samples from the pseudolabelled data. 100% setting thus refers to 3 times the size of real labelled patient data.
> * In general, as seen in table below, as more real labelled data is added performance improves, as generally expected.
>
> |% of real data (pseudolabelled data = 2x real data)|Average AUROC over 3 folds|Average AUPRC over 3 folds|Number of pseudo labelled patient data (fold 0) | Number of real labelled patient data (fold 0)|
> |-|-|-|-|-|
> |25%|0.5326 $\pm$ 0.0152|0.5239 $\pm$ 0.0222|244|122|
> |50%|0.581 $\pm$ 0.0216|0.5505 $\pm$ 0.0274|488|244|
> |75%|0.6888 $\pm$ 0.0257|0.638 $\pm$ 0.0348|732|366|
> |100%|0.7086 $\pm$ 0.0247|0.6533 $\pm$ 0.0374|976|488|
>
> **Q.4: GANDALF vs prior methods**
> * GANDALF differs from WISER and CODE-AE in terms of the strategy used for training.
> * GANDALF aims to augment existing patient data by transforming labelled cell lines, while WISER tries to assign labels to unlabelled patients based on labels available in cell lines.
> * CODE-AE tries to augment the data in the shared domain invariant space.
> * In addition, both WISER and CODE-AE ignore the differences in response distributions across patients and cell lines, which GANDALF addresses via multi-task learning, where cell line response prediction and patient response prediction are treated as two separate tasks.
> * In addition, the evaluation of CODE-AE and WISER in the original papers are on gene expression, whereas our evaluation is on mutation data, which is of clinical relevance as shown in prior literature[1].
>
> **References**
>
> [1] Jayagopal, A., Xue, H., He, Z., Walsh, R. J., Hariprasannan, K. K., Tan, D. S. P., ... & Rajan, V. (2024, August). Personalised Drug Identifier for Cancer Treatment with Transformers using Auxiliary Information. In Proceedings of the 30th ACM SIGKDD Conference on Knowledge Discovery and Data Mining (pp. 5138-5149).

---

### Official Review · Reviewer_w5ow · 2024-10-31

**Soundness:** 4
**Presentation:** 4
**Contribution:** 3
**Rating:** 8
**Confidence:** 3

**Summary:**

Edit: Updated my score to 8 after the authors addressed my most major concerns

This paper proposes a method, GANDALF, for improving domain transfer for machine learning models, explicitly focusing on the cancer domain where we want to train chemo response models on cell line data, but apply them to patients. GANDALF works by creating artificial examples that are more similar to the target domain that can then be used for model training. The authors find that this data augmentation strategy significantly improves the performance of chemo response models on patients.

**Strengths:**

Strengths:
- Very well written paper, I really enjoy the step by step breakdown.
- Very strong empirical improvement over prior art
- Novel combination of techniques, incorporating a wide variety of tricks to improve performance
- Very significant topic, as domain adaptation is critical for many important problems where in-domain data is hard to obtain

**Weaknesses:**

Major:
- This paper is sorta a mix of two contributions, a very good “pseudo-labeler” MTL model and the data augmentation. However, it’s unclear which of those two contributions is important as it’s missing a simple, but important baseline: simply using the “pseudo-labeler” MTL model as a final classifier.

- I don't see any discussion of hyperparmeters for both models and baselines. How were they selected and how did you ensure that equal hyperparameter search time was dedicated to your model and baselines?

- It’s very hard to compare the ablation results in table 2 to table 1. Can you create an alternative table 2 that is side-by-side comparable with table 1?

Minor:
- There are some typos (for example the first row in table 2)
- I think some of the analysis in 3.4 is incorrect. For example, looking at only the top 2 principal components in high dimensional data to compare variance does not seem correct as you might be ignoring a ton of the signal. However, section 3.4 doesn’t matter for the central hypothesis of the paper so it’s not that important. I would actually just remove section 3.4

**Questions:**

Can you explain in more detail the differences between table 2 and 1 and why you chose a vastly different format between them?

How were standard deviations calculated?

---

> ### Author Response · Authors · 2024-11-24
>
> Thank you for your insightful comments and feedback, which have helped us improve our paper. Please find our responses below.
>
> **WEAKNESSES**
>
> **Major W.1: Effect of pseudo-labeler MTL and data augmentation**
> * Thank you for raising this distinction.
> * The “pseudo-labeler” MTL model on its own is similar to the model DruID, where two pretrained domain-specific encoders are attached to a multi-task learning network, trained on both labelled cell lines and patients. Table 1 shows the comparison in performance across DruID and GANDALF.
> * We have also conducted an ablation study removing DDPMs. Our experiments show that DDPMs are important to the overall architecture. We used VAEs instead of DDPMs. We have two domain specific VAEs, and the augmentation is done by passing the latent space representation of cell lines (obtained from the cell line VAE encoder) through the patient VAE decoder. This augmented data is then pseudolabelled as before, using the MTL setup and downstream DRP training is done as in GANDALF. The results, as in the table below (Table 6 in revised paper), show that the use of DDPMs help the model.
>
> |Model|Cis||Flu||Gem||Pac||Tem||
> |-|-|-|-|-|-|-|-|-|-|-|
> ||AUROC|AUPRC|AUROC|AUPRC|AUROC|AUPRC|AUROC|AUPRC|AUROC|AUPRC|
> |GANDALF with tuning|0.6343 $\pm$ 0.0306|0.9093 $\pm$ 0.0355|0.7309 $\pm$ 0.0664|0.8483 $\pm$ 0.0933|0.6188 $\pm$ 0.0674|0.5874 $\pm$ 0.175|0.7728 $\pm$ 0.1253|0.9558 $\pm$ 0.024|0.6451 $\pm$ 0.0776|0.2535 $\pm$ 0.1108|
> |GANDALF W/O DDPM|0.4849 $\pm$ 0.0909|0.8696 $\pm$ 0.0098|0.6929 $\pm$ 0.1189|0.8241 $\pm$ 0.0741|0.5162 $\pm$ 0.1247|0.4932 $\pm$ 0.1867|0.5208 $\pm$ 0.4161|0.8273 $\pm$ 0.1636|0.3138 $\pm$ 0.0647|0.1150 $\pm$ 0.0263|
>
> We have also conducted an additional experiment to evaluate the effect of varying amounts of real and synthetic patient data (Appendix A.3 in revised paper).
>
> **Experiment A:**
> * We retain all the real train patient data and randomly sample 25%, 50%, 75% and 100% of the generated pseudolabelled data, and use this in training the DRP model.
> * 0% setting indicates no augmented data in the DRP training. Results are shown in the table below.
> * 0% does the worst, without any augmentation. Best AUROC is at 50% addition of pseudolabelled data, best AUPRC at 25% pseudolabelled data.
> * Across 25-100% settings, the difference in performance is not statistically significant. For the case of 0% vs any other level of augmentation, differences are statistically significant, indicating that adding psuedolabelled data improves performance. To answer the question of how much pseudolabelled is helpful we will need further studies on possibly larger datasets.
>
> |% of pseudolabelled data|Average AUROC over 3 folds|Average AUPRC over 3 folds|Number of pseudo labelled patient data (fold 0) | Number of real labelled patient data (fold 0)|
> |-|-|-|-|-|
> |0%|0.5263 $\pm$ 0.0195 | 0.5229 $\pm$ 0.0249 |0|488|
> |25%|0.8584 $\pm$ 0.0361|0.7838 $\pm$ 0.0564|15983|488|
> |50%|0.8613 $\pm$ 0.0279|0.7796 $\pm$ 0.0437|31966|488|
> |75%|0.8577 $\pm$ 0.0269|0.7677 $\pm$ 0.0354|47948|488|
> |100%|0.8409 $\pm$ 0.0437|0.778 $\pm$ 0.0255|63931|488|
>
> **Experiment B:**
> * We randomly sample 25%, 50%, 75% and 100% of real labelled patient data. In each case we sample twice the number of real samples from the pseudolabelled data. 100% setting thus refers to 3 times the size of real labelled patient data.
> * In general, as seen in table below, as more real labelled data is added performance improves, as generally expected.
>
> |% of real data (pseudolabelled data = 2x real data)|Average AUROC over 3 folds|Average AUPRC over 3 folds|Number of pseudo labelled patient data (fold 0) | Number of real labelled patient data (fold 0)|
> |-|-|-|-|-|
> |25%|0.5326 $\pm$ 0.0152|0.5239 $\pm$ 0.0222|244|122|
> |50%|0.581 $\pm$ 0.0216|0.5505 $\pm$ 0.0274|488|244|
> |75%|0.6888 $\pm$ 0.0257|0.638 $\pm$ 0.0348|732|366|
> |100%|0.7086 $\pm$ 0.0247|0.6533 $\pm$ 0.0374|976|488|

---

> ### Author Response · Authors · 2024-11-24
>
> **Major W.2: Hyperparameters**
>
> Thank you for raising this. The hyperparameter details are available in our code repo, and hence we had not included them in the earlier draft. We have updated this in Appendix A.4 in the revised paper.
> Details of hyperparameters:
> * For baseline models, we used the hyperparameters defined in each of the papers.
> * For GANDALF, we mainly focused on the hyperparameters in the supervised training stages, key being the lower and upper thresholds and learning rate parameters for the DRP and MTL models.
> * We varied the lower threshold between 0.1 to 0.5 and upper threshold from 0.5 to 0.9, with increments done based on quantiles calculated from predicted probability of response after MTL training. This was done for each drug separately.
> * The hidden layers from the VAE were set to 64 dimensions based on our GPU memory restrictions and batch size was 512.
> * For the transformer we used 64 dimensional embeddings, 4 heads and 8 encoder layers.
> * For the cell line VAE, we used 1024, 128 and 64 hidden units and for patient VAE we used 512, 128 and 64 hidden units. Both VAEs used tanh activation.
> * The DDPM uses linear layers with dimensions as the VAE representation size, and uses dropout and ReLU.
> * The MTL network uses 2 linear layers each in embedding drugs, predicting RECIST and predicting AUDRC, with ReLU activation.
> * We did a hyperparameter sweep using Bayesian Optimization for maximum of 15 runs, using WandB.
> * Max epochs were set to 500 for pretraining, 100 for the MTL and DRP training. Early stopping was done using patient validation set pearson correlation as in [1].
>
> ***References:**
>
> [1] Sharifi-Noghabi, H., Harjandi, P. A., Zolotareva, O., Collins, C. C., & Ester, M. (2021). Out-of-distribution generalization from labelled and unlabelled gene expression data for drug response prediction. Nature Machine Intelligence, 3(11), 962-972.

---

> ### Comment · Reviewer_w5ow · 2024-11-24
> **Quick clarification question**
>
> Thank you for the comprehensive response, which I am still in the process of reading through and understanding.
>
> But due to the short response period I wanted to quickly ask a clarification question to better understand your response.
>
> > The “pseudo-labeler” MTL model on its own is similar to the model DruID, where two pretrained domain-specific encoders are attached to a multi-task learning network, trained on both labelled cell lines and patients. Table 1 shows the comparison in performance across DruID and GANDALF.
>
> You say they are similar, but to be clear, they are not the same? At minimum it looks like the hyperparameters are different? (Can you elaborate on the other differences?)

---

> ### Author Response · Authors · 2024-11-24
>
> **Major W.3: Expansion of Table 2 and more ablation studies**
>
> Thank you for raising this. We have expanded out Table 2 (as shown below) based on the model trained, without drug specific tuning.
>
> |Model|Cis||Flu||Gem||Pac||Tem||
> |-|-|-|-|-|-|-|-|-|-|-|
> ||AUROC|AUPRC|AUROC|AUPRC|AUROC|AUPRC|AUROC|AUPRC|AUROC|AUPRC|
> |GANDALF with tuning|0.6343 $\pm$0.0306|0.9093 $\pm$ 0.0355|0.7309 $\pm$ 0.0664|0.8483 $\pm$ 0.0933|0.6188 $\pm$ 0.0674|0.5874 $\pm$ 0.175|0.7728 $\pm$ 0.1253|0.9558 $\pm$ 0.024|0.6451 $\pm$ 0.0776|0.2535$\pm$ 0.1108|
> |GANDALF without tuning|0.4566 $\pm$ 0.0642|0.8610 $\pm$ 0.0239|0.7119 $\pm$ 0.1318|0.8253 $\pm$ 0.1222|0.6620 $\pm$ 0.0765|0.6239 $\pm$ 0.0516|0.6012 $\pm$ 0.3464|0.9022 $\pm$ 0.1016|0.4227 $\pm$ 0.0426|0.1359 $\pm$ 0.0126|
> |W/O MTL|0.3462 $\pm$ 0.1953|0.8183 $\pm$ 0.0786|0.5167 $\pm$ 0.1367|0.7493 $\pm$ 0.126|0.6019 $\pm$ 0.2622|0.6266 $\pm$ 0.1874|0.3313 $\pm$ 0.2325|0.8118 $\pm$ 0.1103|0.7181 $\pm$ 0.0693|0.3540 $\pm$ 0.2374|
> |W/O cross attention|0.3357 $\pm$ 0.1853|0.8147 $\pm$ 0.074|0.4952 $\pm$ 0.1367|0.7424 $\pm$ 0.1255|0.5741 $\pm$ 0.285|0.6134 $\pm$ 0.1979|0.4067 $\pm$ 0.2715|0.8348 $\pm$ 0.1303|0.7028 $\pm$ 0.0697|0.3480 $\pm$ 0.2372|
> |W/O transformer|0.3685 $\pm$ 0.2166|0.8528 $\pm$ 0.0641|0.5167 $\pm$ 0.0536|0.6749 $\pm$ 0.0578|0.4321 $\pm$ 0.1248|0.4599 $\pm$ 0.2099|0.7440 $\pm$ 0.2115|0.9374 $\pm$ .0713|0.3612 $\pm$ 0.0789|0.1305 $\pm$ 0.0392|
> |W perturbation|0.3666 $\pm$ 0.1113|0.8360 $\pm$ 0.0438|0.4643 $\pm$ 0.25|0.6710 $\pm$ 0.1579|0.3897 $\pm$ 0.0802|0.3591 $\pm$ 0.0651|0.5268 $\pm$ 0.334|0.8765 $\pm$ 0.0851|0.5104 $\pm$ 0.0804|0.2317 $\pm$ 0.1476|
> |W/O augmentation|0.3994 $\pm$ 0.1031|0.8567 $\pm$ 0.0312|0.4857 $\pm$ 0.099|0.6860 $\pm$ 0.0365|0.3997 $\pm$ 0.1524|0.3727 $\pm$ 0.1122|0.5923 $\pm$ 0.3288|0.8951 $\pm$ 0.0812|0.4618 $\pm$ 0.0962|0.2162 $\pm$ 0.1373|
> |W majority vote|0.4780 $\pm$ 0.1499|0.8594 $\pm$ 0.0723|0.6357 $\pm$ 0.118|0.7976 $\pm$ 0.0931|0.3719 $\pm$ 0.1541|0.3575 $\pm$ 0.0535|0.3780 $\pm$ 0.3274|0.7886 $\pm$ 0.1763|0.3672 $\pm$ 0.2096|0.1307 $\pm$ 0.0253|
>
> Expanded version (above) does not tell us much of the contribution of each component, since the ablation experiments were conducted by removing each component cumulatively. This means that the sequence of removal of components also affects the performance. To understand the contribution of each individual component, we added additional ablation studies where each test removes just one component of the architecture. We also performed drug specific tuning, as in Table 1 for consistency (Table 6 in revised paper).
>
> |Model|Cis||Flu||Gem||Pac||Tem||
> |-|-|-|-|-|-|-|-|-|-|-|
> ||AUROC|AUPRC|AUROC|AUPRC|AUROC|AUPRC|AUROC|AUPRC|AUROC|AUPRC|
> |GANDALF with tuning|0.6343 $\pm$0.0306|0.9093 $\pm$ 0.0355|0.7309 $\pm$ 0.0664|0.8483 $\pm$ 0.0933|0.6188 $\pm$ 0.0674|0.5874 $\pm$ 0.175|0.7728 $\pm$ 0.1253|0.9558 $\pm$ 0.024|0.6451 $\pm$ 0.0776|0.2535$\pm$ 0.1108|
> |GANDALF W/O MTL|0.3409 $\pm$ 0.219|0.8101 $\pm$ 0.0793|0.5333 $\pm$ 0.075|0.7345 $\pm$ 0.1|0.5587 $\pm$ 0.1787|0.5697 $\pm$ 0.0628|0.2758 $\pm$ 0.1461|0.7582 $\pm$ 0.1012|0.7513 $\pm$ 0.0805|0.3215 $\pm$ 0.1623|
> |GANDALF W/O cross attention|0.6061 $\pm$ 0.0475|0.9183 $\pm$ 0.0255|0.7309 $\pm$ 0.0834|0.8483 $\pm$ 0.0967|0.6188 $\pm$ 0.0674|0.5873 $\pm$ 0.1753|0.7728 $\pm$ 0.1253|0.9558 $\pm$ 0.024|0.6152 $\pm$ 0.1074|0.2068 $\pm$ 0.0585|
> |GANDALF W/O transformer|0.3735 $\pm$ 0.1404|0.8047 $\pm$ 0.0417|0.4143 $\pm$ 0.1122|0.6400 $\pm$ 0.1231|0.5718 $\pm$ 0.0805|0.4760 $\pm$ 0.0865|0.5625 $\pm$ 0.3903|0.8478 $\pm$ 0.1374|0.2106 $\pm$ 0.0457|0.0993 $\pm$ 0.0195|
> |GANDALF W/O VAE| GPU memory issues||||||||||
> |GANDALF W/O DDPM|0.4849 $\pm$ 0.0909|0.8696 $\pm$ 0.0098|0.6929 $\pm$ 0.1189|0.8241 $\pm$ 0.0741|0.5162 $\pm$ 0.1247|0.4932 $\pm$ 0.1867|0.5208 $\pm$ 0.4161|0.8273 $\pm$ 0.1636|0.3138 $\pm$ 0.0647|0.1150 $\pm$ 0.0263|

---

> ### Author Response · Authors · 2024-11-24
>
> **Minor W.1: Typos**
> Thank you! Will correct (extra “0.”)
>
> **Minor W.2: Use of PCA**
> Thank you for raising this. We agree that some signal is lost in the use of PCA. We only compare the first two principal components across all three datasets. We agree that this is not very important and have moved it to the appendix A.6.
>
> **QUESTIONS**
>
> **Q.1: Table 2 vs Table 1**
> * The results in Table 2 include 7 drugs and obtains AUROC and AUPRC over all of them in one go as in some of the prior papers in DRP. We followed this format to make the presentation of results easier.
> * Following your comments we have included additional tables (added in the appendix Table 6) in a format consistent with Table 1.
>
> **Standard deviation calculation**
> Standard deviation is calculated over 3 test folds.

---

> ### Author Response · Authors · 2024-11-24
> **Response to clarification question**
>
> Please find our response below regarding the differences between DruID and the MTL part of GANDALF.
>
> 1. Yes, the two architectures are not exactly the same. As you rightly identified, the hyperparameters used in each architecture are chosen based on the validation data performance.
> 2. The architecture of DruID differs from the MTL network of GANDALF in the following ways:
>
> (1) DruID takes as input a 7776 dimensional vector (obtained by aggregating information from multiple mutations) which is directly fed into a VAE based encoder-decoder setup, while the MTL network in GANDALF uses a transformer encoder[1] followed by VAE layers.
>
> (2) DruID pretrains two domain specific VAEs using a CORAL loss, while in the MTL network of GANDALF we have a CORAL loss term in the MTL phase as well.
>
> [1] Jayagopal, A., Xue, H., He, Z., Walsh, R. J., Hariprasannan, K. K., Tan, D. S. P., ... & Rajan, V. (2024, August). Personalised Drug Identifier for Cancer Treatment with Transformers using Auxiliary Information. In Proceedings of the 30th ACM SIGKDD Conference on Knowledge Discovery and Data Mining (pp. 5138-5149).

---

> ### Comment · Reviewer_w5ow · 2024-11-24
> **Quick followup question**
>
> Thank you for the quick followup, but I'm a bit confused by your response.
>
> > As you rightly identified, the hyperparameters used in each architecture are chosen based on the validation data performance
>
> If I understand your discussion of hyperparameter selection correctly, this is not quite complete.
>
> It appears in your earlier response that you did not do hyperameter selection for baselines (relying on the selection those papers must have done)
>
> > For baseline models, we used the hyperparameters defined in each of the papers.
>
> This sounds like you only did hyperparameter selection based on **your validation data** for **your model**.
>
> Presumably, DruID was tuned, but it's unclear if they did as comprehensive a search as you + the factor that the tuning was done on their dataset, not yours.
>
> I'm a bit concerned about this experimental procedure. How do you know that their hyperparameters would work well for your data? Looking briefly through the data, it looks like you are using a substantially different dataset from them (it looks like at least double the number of CCLE samples, and major differences in patient samples).

---

> > ### Author Response · Authors · 2024-11-24
> > **Response to followup question**
> >
> > * Yes, the datasets used in the original DruID paper and in GANDALF are different. To ensure fair comparison across all baselines, we first selected the hyperparameters  (like learning rate, hidden dimensions of networks, batch size etc) used in the respective architecture either from the paper or the code base, and used the same range of hyperparameters as is used in these papers.
> > * We next ran hyperparameter sweeps using these ranges of hyperparameters (rather than choosing the exact hyperparameter values of the previous literature), to determine the best hyperparameters for our dataset. The sweeps ran for the same number of times and used a Bayesian optimization strategy across the models.
> > * The results on the test datasets were then used for comparing GANDALF against previous papers, after choosing the best hyperparameters in each case.
> >
> > Hope this clarifies the hyperparameter selection process. Please let us know if any further clarification is needed.

---

> > > ### Comment · Reviewer_w5ow · 2024-11-24
> > > **Still slightly confused**
> > >
> > > I'm sorry, but I'm still confused.
> > >
> > > Can you explicitly list the hyperparameter grid you used for DruID? The DruID paper doesn't seem to specify a hyperparameter search range (instead only listing the final hyperparameters), so where did you get your ranges for that baseline?

---

> > > > ### Author Response · Authors · 2024-11-25
> > > > **Hyperparameters for baselines**
> > > >
> > > > Please find below the details of the hyperparameters we used for the baselines. Hope this clarifies things.
> > > >
> > > > We did not tune epochs since we had early stopping in all cases. Across methods, we focused on the last stage of DRP for tuning.
> > > > * **DruID:** MTL learning rate range [0.001, 0.05], RECIST prediction dimensions for 1st hidden layer - {64, 32}, dimensions for second hidden layer {16, 8}. We took the CCLE-TCGA setting in the original paper. In this particular case, we obtained the ranges from the authors. AUDRC + RECIST learning rate seemed to help the most, along with the number of hidden neurons in the RECIST prediction head. So we added these to the hyperparameter list. We do acknowledge that this may be different from the original paper, but we have tried to ensure a fair comparison.
> > > > * In models like PANCDR, CODE-AE, PREDICT-AI, we used ranges from the respective paper. For example,
> > > >
> > > > **PREDICT-AI:** lr  [0.0001, 0.001], batch size {128, 64}
> > > >
> > > > **PANCDR:** Encoder bottleneck dimensions {100, 128, 256}, GCN d_dim {100, 128, 256}, learning rate [0.0001, 0.001], lr_adv [0.0001, 0.001], lam {1, 0.1, 0.01}, batch size [128, 256]
> > > >
> > > > **CODE-AE:** Dropout {0, 1}. Other tunable parameter was epochs.
> > > >
> > > > **WISER:** Dropout {0, 1}, learning rate [0.001, 0.1]. Inverse temperature was not used in the final DRP phase. Other tunable parameter was epochs.

---

> ### Comment · Reviewer_w5ow · 2024-11-25
> **Thank you for for updates, but I think my main conclusion is unchanged**
>
> Dear authors,
>
> Thank you very much for the additional details and clarification. I think your work is now much more clear and weaknesses 2/3 have been fully addressed. The improved table 2 in particular is a great help for understanding.
>
> However, I don't think I can recommend acceptance here.
>
> The problem is that weakness 1, the lack of experiments showing the value of data augmentation is still very much unresolved. This is especially problematic because data augmentation is the central hypothesis of the paper. We simply have no way of knowing whether the improved performance here is due to data augmentation or due to the author's changes to the MTL model.
>
> And we know those changes to the MTL model were significant. As shown by the author's new table 2 showing the impact of the added transformer layers.
>
> The authors have proposed the DruID vs GANDALF experiment to answer this question, but I don't think that's a satisfactory test since there are so many changes between DruID and GANDALF. Especially since this question is so fundamental to the central hypothesis of the question. I really think this experiment needs to be done correctly by simply testing the performance of the author's new MTL model alone (ignoring any data augmentation additions).
>
> As an aside (and I wish I noticed this in the initial review), but I think there are some concerns about statistical significance here since the standard deviations in table 1 are so high. I think it would be good to explicitly compute p values here to get a better sense of statistical power. My apologies for not pointing this out earlier, and I don't necessarily expect the authors to respond and I have not taken this into account for my score.

---

> > ### Author Response · Authors · 2024-11-26
> >
> > **Without pseudolabelling**
> > We have added an experiment where we used the MTL stage to run evaluation on the test patients without using the pseudolabelled patient data. We hope this alleviates your concern.
> >
> > Please note that the pseudolabelled data in GANDALF is generated in step 4 of the architecture and only used in step 5, while training a separate feedforward DRP network. It is not used to train any of the other stages of the network. As such, to remove the influence of pseudolabelled data, we conducted an experiment where we directly use the MTL part of the network (after stage 3) to run inference on the test patient data (results in Table below).
> >
> > As seen in the table, the average performance drops in all drugs (except AUPRC in Cis and Tem), indicating the benefit of the pseudolabelling step in training.
> >
> > | AUROC            | Cis                          | Flu                          | Gem                        | Pac                         | Tem                        |
> > | -------------------- | ------------------------- | ------------------------- | -------------------------- | ------------------------- | ------------------------- |
> > | GANDALF        | 0.6343 +- 0.0306    | 0.7309 +- 0.0664    | 0.6188 +- 0.0674    | 0.7728 +- 0.1253    | 0.6451 +- 0.0776    |
> > | Without pseudolabelling | 0.6048 +- 0.1185 | 0.6452 +- 0.2304 | 0.6019 +- 0.1891 | 0.6825 +- 0.3345 | 0.5026 +- 0.1647 |
> >
> > | AUPRC            | Cis                          | Flu                          | Gem                        | Pac                         | Tem                        |
> > | -------------------- | ------------------------- | ------------------------- | -------------------------- | ------------------------- | ------------------------- |
> > | GANDALF        | 0.9093 +- 0.0355    | 0.8483 +- 0.0933    | 0.5874 +- 0.175    | 0.9558 +- 0.024    | 0.2535 +- 0.1108    |
> > | Without pseudolabelling | 0.919 +- 0.035 | 0.8146 +- 0.1368 | 0.5669 +- 0.1321 | 0.9066 +- 0.1301 | 0.2702 +- 0.1045 |

---

> > > ### Comment · Reviewer_w5ow · 2024-11-26
> > > **Thank you, updated score**
> > >
> > > Dear authors,
> > >
> > > Thank you for completing that requested experiment. I have updated my score accordingly to accept as my most important concern has been addressed.
> > >
> > > The high variances in your result tables are still a bit concerning, but that should hopefully be tackled in your next work.

---

> > > > ### Author Response · Authors · 2024-11-26
> > > >
> > > > Thanks a lot for your valuable feedback and review, which have helped us improve our paper.

---

### Official Review · Reviewer_64hx · 2024-11-02

**Soundness:** 3
**Presentation:** 2
**Contribution:** 3
**Rating:** 6
**Confidence:** 3

**Summary:**

This study presents GANDALF, a generative, semi-supervised framework addressing the challenge of limited labeled data in sparse patient genomic datasets by leveraging cell line data and augmenting patient samples with their clinical information. However, key concerns include the need for clearer model justification, additional technical details, and further clarity in experimental results.

**Strengths:**

- This study introduces GANDALF, a novel generative and semi-supervised data augmentation framework designed to overcome sparse labeled patient genomic data.
- GANDALF leverages labeled samples from the cell line to augment patient data with critical information.
- By incorporating attention mechanisms and generating augmented samples, GANDALF significantly enhances classifier performance in predicting patient drug response.

**Weaknesses:**

- The manuscript proposes a model with three distinct encoders, but the reasoning behind selecting this multi-encoder structure lacks clarity and strong justification.
- Some technical details are missing.
- Experimental results are not comprehensive.

**Questions:**

- The manuscript introduces a model with three distinct encoders, yet the rationale behind this multi-encoder structure lacks clarity and a convincing explanation. For example, the authors state, "to ease training, we reduce the dimensionality … using variational autoencoders (VAEs),” but it is unclear why a VAE is required rather than a simpler embedding without a decoder and what specific role the VAE’s decoder serves. While VAEs are known for capturing latent features and supporting generative tasks, considering that DDPM is an autoencoder-structured generative model, the need for a secondary, nested autoencoder is not well justified. This makes the model more complicated. The authors are needed to address the following points; 1. why a simpler embedding without a decoder is insufficient in this context; 2. what the specific function of the VAE's decoder plays in the model's architecture; 3. how the combination of VAEs and DDPM provides benefits that DDPM alone does not.

- How is the time step $t$ defined? Does it mean the time after treatment? This should be clarified in the problem formulation section. Additionally, how is $\beta_t$ set?

- How are the ground truth noise distributions $e_c, e_p$ defined, and how does the DDPM decoder remove domain-specific noise?

- In the results section, only five drugs are reported out of about 100 test drugs. The criteria for selecting these specific drugs should be clarified, especially given that all drugs were included in the ablation study.

- The study should assess the model’s ability to accurately label mutation-drug pairs when the drug in question is absent from the labeled training dataset (X_p, y_p, d_p). Were these five drugs excluded from the training and validation sets? If they were, a comparative analysis of model performance on drugs within and outside the training set would enhance the understanding of the model’s generalizability; if not, this experiment does not adequately validate the model’s intended benefits.

- There are typos in Equation (6), specifically where L_MSE is indicated as L_R and L_KLDV as L_KLD?

---

> ### Author Response · Authors · 2024-11-24
>
> Thank you for your questions and comments. We have tried to address the weaknesses listed in detail, in each of the questions raised. Please let us know if any further clarification is needed.
>
> **Q.1 Rationale for VAEs with DDPMs**
> * The use of VAE provides us two main benefits - (1) reduces dimensionality and helps reduce GPU memory used and (2) learns a latent space in a generative manner unlike autoencoders.
>
> **1. Why is a simpler embedding without decoder insufficient?**
> * One of the key challenges in our problem setting is the limited number of samples. As such, we rely on unsupervised pretraining to first learn the latent space. This training uses standard unsupervised training procedure, where a decoder learns to reconstruct the input data. The decoder is thus useful in pretraining.
>
> **2. What is the specific function of the VAE decoder?**
> * We use VAEs specifically for their generative ability. The VAE decoder helps learn the data distribution, unlike standard autoencoders and lends generative power to GANDALF.
> * While DDPMs are also generative models, in GANDALF they help perform domain-specific denoising. Our experiments show that DDPMs are important to the overall architecture. We conducted an ablation study (GANDALF W/O DDPM) where we removed the DDPM layer and instead used VAEs. We have two domain specific VAEs, and the augmentation is done by passing the latent space representation of cell lines (obtained from the cell line VAE encoder) through the patient VAE decoder. This augmented data is then pseudolabelled as before, using the MTL setup and downstream DRP training is done as in GANDALF. The results, as in the table below (also in Table 6 of revised paper), show empirically that the use of DDPMs help the model.
>
> |Model|Cis||Flu||Gem||Pac||Tem||
> |-|-|-|-|-|-|-|-|-|-|-|
> ||AUROC|AUPRC|AUROC|AUPRC|AUROC|AUPRC|AUROC|AUPRC|AUROC|AUPRC|
> |GANDALF with tuning|0.6343 $\pm$ 0.0306|0.9093 $\pm$ 0.0355|0.7309 $\pm$ 0.0664|0.8483 $\pm$ 0.0933|0.6188 $\pm$ 0.0674|0.5874 $\pm$ 0.175|0.7728 $\pm$ 0.1253|0.9558 $\pm$ 0.024|0.6451 $\pm$ 0.0776|0.2535 $\pm$ 0.1108|
> |GANDALF W/O DDPM|0.4849 $\pm$ 0.0909|0.8696 $\pm$ 0.0098|0.6929 $\pm$ 0.1189|0.8241 $\pm$ 0.0741|0.5162 $\pm$ 0.1247|0.4932 $\pm$ 0.1867|0.5208 $\pm$ 0.4161|0.8273 $\pm$ 0.1636|0.3138 $\pm$ 0.0647|0.1150 $\pm$ 0.0263|
>
> **3. How does the combination of VAE and DDPM help?**
> * The combination of VAE and DDPM enables us to work on a lower dimensional latent space and helps reduce compute requirements, akin to latent diffusion models[1]. We do observe an out of memory issue (Table 6 in revised paper) when we bypassed the VAE and directly passed the output of the transformer encoder to the DDPM layer.
>
> **Q.2 Definitions of** $t$ **and** $\beta_t$
> * $t$ corresponds to number of DDPM diffusion steps (length of Markov chain) - we have added this to the paper in the problem formulation (Line 303)
> * Beta generation is based on the cosine beta scheduling proposed by [2].
> $$\beta_t = 1 - \frac{\bar{\alpha_t}}{\bar{\alpha_{t-1}}}$$
> $$\bar{\alpha_t} = \frac{f(t)}{f(0)}$$
> $$f(t) = cos(\frac{t + s}{1+s}.\frac{\pi}{2})^2$$
> T is the number of diffusion steps, s= 0.008 as in [2], $\beta_t$ is capped to 0.999 as in [2].
>
> **Q.3 Noise definitions and denoising**
> * Ground truth noise is obtained by random sampling from a standard normal distribution (done per batch in training)
> DDPM decoder estimates noise and denoising happens via eq 5 in the paper. During decoding the DDPM estimates the noise separately for the two domains and the loss term in eq. 3 ensures that each decoder learns to remove the domain-specific noise. This formulation is similar to that in [3].
>
> $$L_{DDPM} = E_{(Z_{vc}, \epsilon_c, t)}[\epsilon_c - \epsilon_{c\theta}(Z_{vct}, t)]^2 + E_{(Z_{vp}, \epsilon_p, t)}[\epsilon_p - \epsilon_{p\theta}(Z_{vpt}, t)]^2$$
>
> $$X_c = denoise(Z_{vct}, t, \epsilon_{c\theta})$$
>
> $$X_p = denoise(Z_{vpt}, t, \epsilon_{p\theta})$$
>
> $$denoise(X_t, t, \epsilon) = \frac{1}{\sqrt{\hat{\alpha_{t}}}}(X_t - \sqrt{1-\hat{\alpha_{t}}}\epsilon); \hat{\alpha_{t}} = \Pi_{i=1}^{t}(\alpha_i); \alpha_t = 1 - \beta_t$$
>
>
> **References:**
>
> [1] Rombach, R., Blattmann, A., Lorenz, D., Esser, P., & Ommer, B. (2022). High-resolution image synthesis with latent diffusion models. In Proceedings of the IEEE/CVF conference on computer vision and pattern recognition (pp. 10684-10695).
>
> [2] Nichol, A. Q., & Dhariwal, P. (2021, July). Improved denoising diffusion probabilistic models. In International conference on machine learning (pp. 8162-8171). PMLR.
>
> [3] Ho, J., Jain, A., & Abbeel, P. (2020). Denoising diffusion probabilistic models. Advances in neural information processing systems, 33, 6840-6851.

---

> ### Author Response · Authors · 2024-11-24
>
> **Q.4 Drug selection criteria**
> * We would like to clarify that the ablation study includes 7 drugs, not all the 56 drugs mentioned earlier. Please find below the detailed steps we used to decide on the 5 drugs mentioned in Table 1.
> * Details of the filtering criteria (reference Appendix A.1.1 in revised paper):
> 1. For each of the 56 drugs in patients, we first consider those with at least 20 labelled patient samples[1] - this reduced labelled data to 12 drugs
> 2. For each drug, we divided the samples into groups based on cancer type and data source. Each group with > 20 samples was divided into ⅔-⅓ ratio in 3-fold label based stratified CV. For some groups, no test samples were available. We excluded these to get 7 drugs
> 3. We removed drugs which had < 3 positive samples as it would cause issues in CV, where one fold may have test samples with only a single label - this resulted in the five drugs shown in Table 1
> 4. The 7 drugs in step (2) above were used in the ablation studies in Table 2, to report overall performance metrics.
>
> **Q.5 Generalization to other drugs**
> * We apologize for the confusion. In this paper, our focus is on predicting the response of patients to a standard set of chemotherapy drugs (which is also a challenging task as seen in prior literature), which are used in clinical practice. We did not focus on drug repurposing, where we need to generalize to drugs outside of the training set. We have edited the problem formulation to make this more clear (Section 3.1 in revised paper).
> * We follow the approach taken in prior DRP methods[1, 2, 3] to predict the response of patients to each individual drug, where sufficient labelled samples are available. We build drug-specific models, as in [1, 2, 3] while incorporating drug information (via Morgan fingerprints).
> * The main challenge in predicting patient drug response is the limited availability of labelled patient data.
> * Number of drugs with a labelled response in patients is also not a uniform distribution, i.e. some drugs may have more patients with documented response (Cisplatin has ~100 patients) while others may have fewer (Oxaliplatin has < 5 labelled patients). This prevents a comprehensive evaluation across all available drugs/ broader drug set at this juncture.
>
> **Q 6: Typos and loss terms**
> We intended to reuse the same equations, we have redefined them in the paper to add clarity (Eq 7 in revised paper).
>
> **References:**
>
> [1] He, D., Liu, Q., Wu, Y., & Xie, L. (2022). A context-aware deconfounding autoencoder for robust prediction of personalized clinical drug response from cell-line compound screening. Nature Machine Intelligence, 4(10), 879-892.
>
> [2] Ma, J., Fong, S. H., Luo, Y., Bakkenist, C. J., Shen, J. P., Mourragui, S., ... & Ideker, T. (2021). Few-shot learning creates predictive models of drug response that translate from high-throughput screens to individual patients. Nature Cancer, 2(2), 233-244.
>
> [3] Peres da Silva, R., Suphavilai, C., & Nagarajan, N. (2021). TUGDA: task uncertainty guided domain adaptation for robust generalization of cancer drug response prediction from in vitro to in vivo settings. Bioinformatics, 37(Supplement_1), i76-i83.

---

> > ### Author Response · Authors · 2024-11-25
> >
> > We hope you have had a chance to look at the revised version of our paper, where we have included changes as per the feedback we received. Thank you for your comments which have helped us improve the paper. Please let us know if there is any additional information we can provide. Thank you!

---

> > > ### Comment · Reviewer_64hx · 2024-11-25
> > >
> > > Thanks for your reply. I've raised my score accordingly.

---

> > > > ### Author Response · Authors · 2024-11-26
> > > >
> > > > Thank you very much for the review and feedback!

---

> > > > ### Author Response · Authors · 2024-11-28
> > > >
> > > > With the extended discussion window, if there is any other clarification you need from us, please let us know. Thank you once again for your review.

---

### Official Review · Reviewer_RAe2 · 2024-11-04

**Soundness:** 2
**Presentation:** 2
**Contribution:** 2
**Rating:** 6
**Confidence:** 3

**Summary:**

The authors propose GANDALF, a framework that leverages attention-based generative models to augment patient genomic data, addressing limitations in data availability and patient-specific characteristics within existing models. GANDALF demonstrates superior performance over state-of-the-art models in predicting drug responses for cancer treatments.

**Strengths:**

- The studied problem is important and practical.

- The idea of incorporating patient-specific characteristics with drug response prediction models is novel.

- The code is provided for reproducibility.

**Weaknesses:**

- The comparative performance analysis in Table 1 focuses on only 5 drugs, which is a small subset of the 56 drugs mentioned in the dataset. The paper does not justify why these specific drugs were chosen or demonstrate that the performance advantages generalize across the broader drug set. This limited evaluation makes it difficult to assess the method's robustness across different drug classes.  The paper needs to demonstrate results across a more diverse set of drug classes and provide statistical tests showing whether performance advantages hold across the full drug set.

- The authors state "we do drug-specific model tuning in GANDALF, by only augmenting with sample, drug pairs for the drug considered." This approach may cause extensive computational burden, as it requires fine-tuning for each new drug, potentially limiting its practical deployment.

- The paper does not address how well GANDALF generalizes to novel drugs in out-of-domain scenarios—a critical consideration given that new drugs are regularly introduced into clinical practice.

- Though the paper deals with the challenge of limited patient data, it doesn't systematically analyze how performance varies with different amounts of training data. There's no exploration of learning curves showing how model performance changes with increasing patient samples, or minimum data requirements for reliable predictions.

**Questions:**

See weaknesses above.

---

> ### Author Response · Authors · 2024-11-24
>
> Thank you for your valuable feedback and comments, which have helped us improve our paper. Please find our responses below.
>
> **W.1 Evaluation on a broader drug set**
> * Thank you for raising this question.
> * First, we would like to clarify that we are interested in predicting responses for specific drugs which are clinically approved for use in cancer. Typically this is a small number (100-300). Such models can be used in clinical decision support systems for personalized treatment planning by doctors. This is different from a drug repurposing model which can predict responses to new and unseen drugs, and has use cases in other contexts as well, such as drug discovery.
> * The main challenge in predicting patient drug response is the limited availability of labelled patient data.
> * Number of drugs with a labelled response in patients is also not uniformly distributed, i.e. some drugs may have more patients with documented response (Cisplatin has ~100 patients in our dataset) while others may have fewer (Oxaliplatin has < 5 labelled patients). This prevents a comprehensive evaluation across all available drugs/ broader drug set.
> * To evaluate our model, we follow the approach taken by prior DRP papers [1, 2, 3] to predict the response of patients to each individual drug, where sufficient labelled samples are available. We build drug-specific models, as in [1, 2, 3] while incorporating drug information in the model.
> * Specifically, we evaluate our model on drugs with > 20 labelled samples for train-test split creation, as in [1].
> * Details of the filtering criteria (reference Appendix A.1.1 in updated paper):
> 1. For each of the 56 drugs in patients, we first consider those with at least 20 labelled patient samples[1] - this reduced labelled data to 12 drugs
> 2. For each drug, we divided the samples into groups based on cancer type and data source. Each group with > 20 samples was divided into ⅔-⅓ ratio in 3-fold label based stratified CV. For some groups, no test samples were available. We excluded these to get 7 drugs
> 3. We removed drugs which had < 3 positive samples as it would cause issues in CV, where one fold may have test samples with only a single label - this resulted in the five drugs shown in Table 1
> 4. The 7 drugs in step (2) above were used in the ablation studies in Table 2, to report overall performance metrics.
>
> **W.2 Drug-specific fine tuning**
> * We would like to clarify that drug specific tuning here refers to using only a subset of the pseudolabelled (sample, drug) pairs, where drug is the drug of interest.
> * This way of developing drug-specific models is the norm in DRP literature [1, 2, 3] and we follow the same. We do acknowledge that the time complexity for training would be in O(D) where D is the number of available drugs.
> * In training GANDALF, the tuning is done only in Step 5 - training the DRP classifier.
> * Additionally, as discussed above, our focus in this paper is to predict drug response for a limited set of clinically approved drugs, not to allow drug repurposing with new drugs.
>
> **W.3 Out-of-domain drug generalization**
> * We would like to clarify that we are not performing drug repurposing, i.e. generalizing to out of domain scenarios.
> * We are interested in predicting responses to an existing set of clinically approved drugs (100-300), which is still a challenging task as discussed above.
> * However, this is definitely a potential area of further research, and can serve as extensions of our current work.
>
> **References:**
>
> [1] He, D., Liu, Q., Wu, Y., & Xie, L. (2022). A context-aware deconfounding autoencoder for robust prediction of personalized clinical drug response from cell-line compound screening. Nature Machine Intelligence, 4(10), 879-892.
>
> [2] Ma, J., Fong, S. H., Luo, Y., Bakkenist, C. J., Shen, J. P., Mourragui, S., ... & Ideker, T. (2021). Few-shot learning creates predictive models of drug response that translate from high-throughput screens to individual patients. Nature Cancer, 2(2), 233-244.
>
> [3] Peres da Silva, R., Suphavilai, C., & Nagarajan, N. (2021). TUGDA: task uncertainty guided domain adaptation for robust generalization of cancer drug response prediction from in vitro to in vivo settings. Bioinformatics, 37(Supplement_1), i76-i83.

---

> ### Author Response · Authors · 2024-11-24
>
> **W.4 Effect of different amounts of training data**
> * Thank you for raising this.
> * We had included learning curves corresponding to two data augmentation strategies in Appendix Figure 6 (in the updated paper) - (1) no data augmentation using only patient data and (2) doubling the size of labelled patient data by adding Gaussian noise (details in section 3.5). In addition, we have now added the learning curve for GANDALF based data augmentation.
> * We have now conducted an additional experiment to evaluate the effect of varying amounts of real and synthetic patient data (Appendix A.3 in the updated paper).
>
> **Experiment A:**
> * We retain all the real train patient data and randomly sample 25%, 50%, 75% and 100% of the generated pseudolabelled data, and use this in training the DRP model.
> * 0% setting indicates no augmented data in the DRP training. Results are shown in the table below (Table 4 in revised paper).
> * 0% does the worst, without any augmentation. Best AUROC is at 50% addition of pseudolabelled data, best AUPRC at 25% pseudolabelled data.
> * Across 25-100% settings, the difference in performance is not statistically significant. For the case of 0% vs any other level of augmentation, differences are statistically significant, indicating that adding psuedolabelled data improves performance. To answer the question of how much pseudolabelled is helpful we will need further studies on possibly larger datasets.
>
> |% of pseudolabelled data|Average AUROC over 3 folds|Average AUPRC over 3 folds|Number of pseudo labelled patient data (fold 0) | Number of real labelled patient data (fold 0)|
> |-|-|-|-|-|
> |0%|0.5263 $\pm$ 0.0195 | 0.5229 $\pm$ 0.0249 |0|488|
> |25%|0.8584 $\pm$ 0.0361|0.7838 $\pm$ 0.0564|15983|488|
> |50%|0.8613 $\pm$ 0.0279|0.7796 $\pm$ 0.0437|31966|488|
> |75%|0.8577 $\pm$ 0.0269|0.7677 $\pm$ 0.0354|47948|488|
> |100%|0.8409 $\pm$ 0.0437|0.778 $\pm$ 0.0255|63931|488|
>
> **Experiment B:**
> * We randomly sample 25%, 50%, 75% and 100% of real labelled patient data. In each case we sample twice the number of real samples from the pseudolabelled data. 100% setting thus refers to 3 times the size of real labelled patient data.
> * In general, as seen in table below (Table 5 in revised paper), as more real labelled data is added performance improves, as generally expected.
>
> |% of real data (pseudolabelled data = 2x real data)|Average AUROC over 3 folds|Average AUPRC over 3 folds|Number of pseudo labelled patient data (fold 0) | Number of real labelled patient data (fold 0)|
> |-|-|-|-|-|
> |25%|0.5326 $\pm$ 0.0152|0.5239 $\pm$ 0.0222|244|122|
> |50%|0.581 $\pm$ 0.0216|0.5505 $\pm$ 0.0274|488|244|
> |75%|0.6888 $\pm$ 0.0257|0.638 $\pm$ 0.0348|732|366|
> |100%|0.7086 $\pm$ 0.0247|0.6533 $\pm$ 0.0374|976|488|
>
> We hope we have addressed all your questions listed above. Please let us know if any further clarification is needed.

---

> > ### Author Response · Authors · 2024-11-25
> >
> > Thank you for your feedback and review. We hope you have had a chance to look at our revised paper, with edits made as per your feedback. Please let us know if you would like any additional information from our end. Thank you!

---

> > > ### Comment · Reviewer_RAe2 · 2024-11-25
> > >
> > > Thanks for your response. I have increased my score accordingly.

---

> > > > ### Author Response · Authors · 2024-11-28
> > > >
> > > > With the extended discussion window, if there is any other clarification you need from us, please let us know. Thank you once again for your review.

---

> ### Author Response · Authors · 2024-11-26
>
> Thank you very much for your review and feedback!

---

### Author Response · Authors · 2024-12-02

We **sincerely thank all the reviewers** for their valuable feedback and comments. We have addressed all the questions and concerns raised and updated the paper accordingly.

New results and clarifications in our rebuttal include:

* Elaboration on the choice of drugs used in the evaluation
* Experiments to examine the effect of varying amounts of training data on performance (both real and synthetic)
* Elaboration on the need for VAE, DDPM and transformer in GANDALF architecture.
* Results showcasing the benefits of pseudo labeling and multi task learning network in overall performance improvement
* Clarification on hyperparameters and tuning
* Additional ablation studies involving removal of one component at a time, with drug specific tuning
* Clarification on datasets used
* Addition of TSNE plots

As the discussion period draws to a close, please let us know if we can provide any additional information or clarification. Once again, we are extremely grateful for all the reviews we have received, which have enabled us to improve our paper.

---

### Meta-Review · Area_Chair_vYKz · 2024-12-10

**Metareview:**

This work proposes a new semi-supervised architecture to perform drug response predictions at the patient lavel. They combine attention-based generative models for data augmentation with meta-learning to propose GANDALF, showing improved performance compared to state-of-the-art methods.

The authors have provided additional experiments during the discussion phase, which have addressed the concerns on missing ablations and comparisons. All reviewers now have a positive assessment of the paper.

**Additional Comments On Reviewer Discussion:**

There was no reviewer discussion given the engagement during the reviewer-author discussion, and the consensus reached.

---

### Decision · Program_Chairs · 2025-01-22

Accept (Poster)